# Exploring the Robustness of Distributional Reinforcement Learning against Noisy State Observations

## Abstract

In real scenarios, state observations that an agent observes may contain measurement errors or adversarial noises, misleading the agent to take suboptimal actions or even collapse while training. In this paper, we study the training robustness of distributional Reinforcement Learning (RL), a class of state-of-the-art methods that estimate the whole distribution, as opposed to only the expectation, of the total return. Firstly, we propose State-Noisy Markov Decision Process (SN-MDP) in the tabular case to incorporate both random and adversarial state observation noises, in which the contraction of both expectation-based and distributional Bellman operators is derived. Beyond SN-MDP with the function approximation, we firstly analyze the vulnerability of least squared loss in expectation-based RL, and by contrast theoretically characterize the bounded gradient norm of histogram-based distributional loss, accounting for the better training robustness of distribution RL. Finally, extensive experiments on the suite of games show that in SN-MDP-like setting both expectation-based and distributional RL can converge albeit corresponding to different levels under various state observation noises. However, distributional RL can enjoys better training robustness in the more complicated noisy state observation settings compared with its expectation-based counterpart.

## 1 Introduction

Learning robust and high-performance policies for continuous state-action reinforcement learning (RL) domains is crucial to enable the successful adoption of deep RL in robotics, autonomy, and control problems. However, recent works have demonstrated that deep RL algorithms are vulnerable either to model uncertainties or external disturbances (Huang et al., 2017; Pattanaik et al., 2017; Ilahi et al., 2020; Chen et al., 2019; Zhang et al., 2020; Shen et al., 2020; Singh et al., 2020; Guan et al., 2020). Particularly, model uncertainties normally occur in a noisy reinforcement learning environment where the agent often encounters systematic or stochastic measurement errors on state observations, such as the inexact locations and velocity obtained from the equipped sensors of a robot. On the other hand, external disturbances are normally adversarial in nature. For instance, the adversary can construct adversarial perturbations on state observations to degrade the performance of deep RL algorithms. These two factors lead to noisy state observations that influence the performance of algorithms, precluding the success of reinforcement learning in real-world applications.

Existing works mainly focus on improving the robustness of algorithms in the *test environment* with noisy state observations. Smooth Regularized Reinforcement Learning (Shen et al., 2020) introduced a regularization to enforce smoothness in the learned policy, and thus improved its robustness against measurement errors in the test environment. Similarly, the State-Adversarial Markov decision process (SA-MDP) (Zhang et al., 2020) was proposed and the resulting principled policy regularization enhances the adversarial robustness of various kinds of RL algorithms against adversarial noisy state observations. However, both of these works assumed that the agent can access *clean* state observations during training, which is normally not feasible when the environment is inherently noisy, such as unavoidable measurement errors. Thus, the maintenance and formal analysis of policies robust to noisy state observations during *training* is a worthwhile area of research.

On the other hand, recent distributional reinforcement learning algorithms, including C51 (Bellemare et al., 2017), Quantile-Regression DQN (QRDQN) (Dabney et al., 2018b), Implicit Quantile Networks (Dabney et al., 2018a) and Moment-Matching DQN (MMD) (Nguyen et al., 2020), constantly set new records in Atari games, gaining huge attention in the research community. However, existing literature mainly focuses on the performance of algorithms, other benefits, including the robustness in the noisy environment, of distributional RL algorithms are less studied. As distributional RL can leverage additional information about distribution that captures the uncertainty of the environment more accurately, it is natural to expect that distributional RL with this better representation capability can be less vulnerable to the noisy environment while training, which motivates our research.

In this paper, we investigate the robustness of distributional RL against various kinds of state observation noises encountered during training. Firstly, we propose a general State-Noisy MDP in the tabular setting, in which we prove the convergence of distributional Bellman operator. We further extend SN-MDP to the function approximation case by considering more complex noisy state observations. Notably, we analyze the vulnerability of classical RL and in contrast characterize the Lipschitz continuity *blessing* resulting from the Histogram distributional loss in distributional RL, which leads to a bounded gradient norm. This better behaved gradient mitigates the impact of noisy states on the objective function, accounting for the less vulnerability of distributional RL while training. Finally, extensive experiments demonstrate that both expectation-based and distributional RL algorithms can converge in SN-MDP-like settings. More importantly, distributional RL algorithms tend to achieve better robust performance in the presence of more complex state observation noises compared with its expectation-based counterpart that may even diverge in some cases. These empirical results in Section 5 echo our previous theoretical results in both Section 3 and 4. Overall, the training robustness advantage of distributional RL algorithms we revealed facilitates their deployment especially in the noisy environment.

## 2 BACKGROUND

### 2.1 DISTRIBUTIONAL REINFORCEMENT LEARNING

In the tabular setting without noisy states, the agent's interaction with its environment can be naturally modeled as a standard Markov Decision Process (MDP), a 5-tuple $(\mathcal{S}, \mathcal{A}, R, P, \gamma)$. $\mathcal{S}$ and $\mathcal{A}$ are the state and action spaces, $P : \mathcal{S} \times \mathcal{A} \times \mathcal{S} \to [0, 1]$ is the environment transition dynamics, $R : \mathcal{S} \times \mathcal{A} \times \mathcal{S} \to \mathbb{R}$ is the reward function and $\gamma \in (0, 1)$ is the discount factor.

**Value Function vs Value Distribution.** Firstly, we denote the *return* where $s_t = s$ as $Z^\pi(s) = \sum_{k=0}^\infty \gamma^k r_{t+k+1}$, representing the cumulative rewards following a policy $\pi$, and $r_{t+k+1}$ is reward scalar obtained in the step $t + k + 1$. In the algorithm design, traditional expectation-based RL normally focuses on *value function* $V^\pi(s)$, the expectation of the random variable $Z^\pi(s)$:

$$V^\pi(s) := \mathbb{E}\left[Z^\pi(s)\right] = \mathbb{E}\left[\sum_{k=0}^\infty \gamma^k r_{t+k+1} \mid s_t = s\right]. \tag{1}$$

In contrast, in the distributional RL setting, we focus on the *value distribution*, the full distribution of $Z^\pi(s)$, and the *state-action value distribution* $Z^\pi(s, a)$ in the control problem where $s_t = s, a_t = a$. Both of these distributions can better capture the uncertainty of returns in the MDP beyond just its expectation (Dabney et al., 2018a; Mavrin et al., 2019).

**Distributional Bellman Operator.** In expectation-based RL, we update the value function via the Bellman operator $\mathcal{T}^\pi$, while in distributional RL, the updating is on the value distribution via the *distributional Bellman operator* $\mathfrak{T}^\pi$. To derive $\mathfrak{T}^\pi$, we firstly define the transition operator $P^\pi : \mathcal{Z} \to \mathcal{Z}$:

$$\mathcal{P}^\pi Z(s, a) \overset{D}{:=} Z\left(S', A'\right), S' \sim P(\cdot|s, a), A' \sim \pi\left(\cdot|S'\right), \tag{2}$$

where we use capital letters $S'$ and $A'$ to emphasize the random nature of both, and $\overset{D}{:=}$ indicates convergence in distribution. For simplicity, we denote $Z^\pi(s, a)$ by $Z(s, a)$. Thus, the distributional Bellman operator $\mathfrak{T}^\pi$ is defined as:

$$\mathfrak{T}^\pi Z(s, a) \overset{D}{:=} R(s, a, S') + \gamma \mathcal{P}^\pi Z(s, a). \tag{3}$$

More importantly, $\mathfrak{T}^\pi$ is still a contraction for policy evaluation under the maximal form of the Wasserstein metric $d_p$ over the true and parametric value distributions (Bellemare et al., 2017; Dabney et al., 2018b), where the $p$-Wasserstein metric $d_p$ is defined as

$$d_p = \left( \int_0^1 \left| F_{Z^*}^{-1}(\omega) - F_{Z_\theta}^{-1}(\omega) \right|^p d\omega \right)^{1/p}, \tag{4}$$

which minimizes the distance between the true value distribution $Z^*$ and the parametric distribution $Z_\theta$. $F^{-1}$ is the inverse cumulative distribution function of a random variable with the cumulative distribution function as $F$. In the control setting, the distributional analogue of the Bellman optimality operator converges to the set of optimal value distributions, although it is in a weak sense and requires more involved arguments (Dabney et al., 2018b).

## 2.2 Two Kinds of Noisy State Observations

We investigate both random and adversarial training robustness, *i.e., the performance of RL algorithms under these two types of noisy state observations*, between the expectation-based and distributional RL algorithms. We consider continuous state observations with continuous noises. In the random noisy state case, we apply Gaussian noises with mean 0 and different standard deviations to state features to simulate the measurement error stemming from various sources.

In the adversarial state perturbation setting, we construct white-box adversarial perturbations on state observations for the current policy during training, following the strategy proposed in (Huang et al., 2017; Pattanaik et al., 2017) that leveraged the gradient information of an engineered loss function. In particular, we denote $a_w^t$ as the "worst" action, with the lowest probability from the current policy $\pi_t(a|s)$ in the training step $t$. Thus, the optimal adversarial perturbation $\eta_t$, constrained in an $\epsilon$-ball, can be derived by minimizing the objective function $J$:

$$\min_\eta J(s_t + \eta, \pi_t) = -\sum_{i=1}^n p_i^t \log \pi_t(a_i|s_t + \eta), s.t. \|\eta\| \le \epsilon, \tag{5}$$

where $p_i^t = 1$ if $i$ corresponds to the index of the least-chosen action, i.e. the $w$-th index in the vector $a$, otherwise $p_i^t = 0$. In other words, we construct a target one-hot action $p^t$ with 1 assigned to the index of the least-chosen action. Through this minimization in the form of the cross entropy loss, we can construct the state perturbations $\eta_t$ that can force the policy to choose the least-chosen action $a_w^t$ in each $t$ step.

## 3 Tabular Case: State-Noisy Markov Decision Process

In this section, we extend State-Adversarial MDP (Zhang et al., 2020) to a more general State-Noisy Markov Decision Process (SN-MDP), and particularly provide a proof of the convergence and contraction of distributional Bellman operator in this setting.

### 3.1 Definitions

As shown in Figure 1, SN-MDP is a 6-tuple $(\mathcal{S}, \mathcal{A}, R, P, \gamma, N)$, where the noise generating mechanism $N(\cdot|s)$ maps the state from $s$ to $v(s)$ using either random or adversarial noise with the Marko-

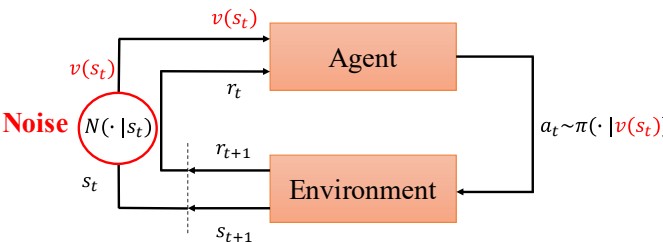

Figure 1: State-Noisy Markov Decision Process where $v(s_t)$ is perturbed by the noise mechanism $N$.

vian and stationary probability $N(v(s)|s)$. It is worthwhile to note that the explicit definition of the noise mechanism $N$ here is based on discrete state transitions, but the analysis can be naturally extended to the continuous case if we let the state space go to infinity. Moreover, let $\mathcal{B}(s)$ be the set that contains the allowed noise space for the noise generating mechanism $N$, i.e., $v(s) \in \mathcal{B}(s)$.

Following the setting in (Zhang et al., 2020), we only manipulate state observations but do not change the underlying environment transition dynamics based on $s$ or the agent's actions directly. As such, our SN-MDP is more suitable to model the random measurement error, e.g., sensor errors and equipment inaccuracies, and adversarial state observation perturbations in safety-critical scenarios.

## 3.2 ANALYSIS OF SN-MDP FOR EXPECTATION-BASED RL

We define the value function $\tilde{V}_{\pi \circ N}$ given $\pi$ in SN-MDP. The Bellman Equations are given by:

$$\tilde{V}_{\pi \circ N}(s) = \sum_a \sum_{v(s)} N(v(s)|s)\pi(a|v(s)) \sum_{s'} p(s'|s,a) \cdot \left[ R(s,a,s') + \gamma \tilde{V}_{\pi \circ N}(s') \right]. \quad (6)$$

The random noise transits $s$ into $v(s)$ with a certain probability and the adversarial noise is the special case of $N(v(s)|s)$ where $N(v^*(s)|s) = 1$ if $v^*(s)$ is the optimal adversarial noisy state given $s$, and $N(v(s)|s) = 0$ otherwise. We denote Bellman operators under random noise mechanism $N^r(\cdot|s)$ and adversarial noise mechanism $N^*(\cdot|s)$ as $\mathcal{T}_r^\pi$ and $\mathcal{T}_a^\pi$, respectively. This implies that $\mathcal{T}_r^\pi \tilde{V}_{\pi \circ N} = \tilde{V}_{\pi \circ N^r}$ and $\mathcal{T}_a^\pi \tilde{V}_{\pi \circ N} = \tilde{V}_{\pi \circ N^*}$. We extend Theorem 1 in (Zhang et al., 2020) to both random and adversarial noise scenario, and immediately obtain that both $\mathcal{T}_r^\pi$ and $\mathcal{T}_a^\pi$ are contraction operators in SN-MDP. We explain this in Theorem 3 of Appendix A.

The pivotal conclusion from Theorem 3 is $\mathcal{T}_a^\pi \tilde{V}_{\pi \circ N} = \min_N \tilde{V}_{\pi \circ N}$. This implies that the adversary attempts to minimize the value function, forcing the agent to select the worse-case action among the allowed transition probability space $N(\cdot|s)$ for each state $s$. The main proof idea is that Bellman updates in SN-MDP result in the convergence to the value function for another "merged" policy $\pi'$ where $\pi'(a|s) = \sum_{v(s)} N(v(s)|s)\pi(a|v(s))$. The value function for the merged policy might ba far away from that for the original policy $\pi$, which tends to worsen the performance of RL algorithms.

## 3.3 ANALYSIS OF SN-MDP IN DISTRIBUTIONAL RL

In the SN-MDP setting for distributional RL, the new distributional Bellman equations use new transition operators in place of $\mathcal{P}^\pi$ in Eq. 2. The new transition operators $\mathcal{P}_r^\pi$ and $\mathcal{P}_a^\pi$, for the random and adversarial settings, are defined as:

$$\mathcal{P}_r^\pi Z_N(s,a) :\overset{D}{=} Z_{N^r}(S',A'), A' \sim \pi(\cdot|V(S')), \text{and } \mathcal{P}_a^\pi Z_N(s,a) :\overset{D}{=} Z_{N^*}(S',A'), A' \sim \pi(\cdot|V^*(S')), \quad (7)$$

where $V(S') \sim N^r(\cdot|S')$ is the state random variable after the transition, and $V^*(S')$ is attained from $N^*(\cdot|S')$ under the optimal adversary. Besides, $S' \sim P(\cdot|s,a)$. Thus, the corresponding new distributional Bellman operators $\mathfrak{T}_r^\pi$ and $\mathfrak{T}_a^\pi$ are:

$$\mathfrak{T}_r^\pi Z_N(s,a) :\overset{D}{=} R(s,a,S') + \gamma \mathcal{P}_r^\pi Z_N(s,a), \text{and } \mathfrak{T}_a^\pi Z_N(s,a) :\overset{D}{=} R(s,a,S') + \gamma \mathcal{P}_a^\pi Z_N(s,a). \quad (8)$$

In this sense, four sources of randomness define the new compound distribution in the SN-MDP: (1) randomness of reward, (2) randomness in the new environment transition dynamics $\mathcal{P}_r^\pi$ or $\mathcal{P}_a^\pi$ that additionally includes (3) the stochasticity of the noisy transition $N$, and (4) the random next-state value distribution $Z(S',A')$. Besides, the premise of the robustness of distributional RL against noisy state observations lies in the convergence of the new derived distribution Bellman Operators in SN-MDP setting. We proved this convergence and contraction for policy evaluation in Theorem 1.

**Theorem 1.** *(Convergence and Contraction of Distributional Bellman operators in the SN-MDP) Given a policy $\pi$, we define the distributional Bellman operators $\mathfrak{T}_r^\pi$ and $\mathfrak{T}_a^\pi$ in Eq. 8, and consider the Wasserstein metric $d_p$, the following results hold.*

*(1) $\mathfrak{T}_r^\pi$ is a contraction under the maximal form of $d_p$.*

*(2) $\mathfrak{T}_a^\pi$ is also a contraction under the maximal form of $d_p$, following the greedy adversarial rule, i.e., $N^*(\cdot|s') = \arg\min_{N(\cdot|s')} \mathbb{E}[Z(s',a')]$ where $a' \sim \pi(\cdot|V(s'))$ and $V(s') \sim N(\cdot|s')$.*

We provide the proof in Appendix B. The convergence of distributional Bellman operators in the SN-MDP is one of our main contributions. This result allows us to deploy distributional reinforcement learning algorithms comfortably even in settings with noisy state observations.

## 4 FUNCTION APPROXIMATION CASE: NOISY SETTINGS BEYOND SN-MDP

In real scenarios, especially safety-critical cases, perturbations on state observations can be more complicated. For instance, the adversary might perform attacks at certain intervals, yielding unbalanced state observation pairs with a perturbed current state and a benign next state and vice versa. This type of unbalanced perturbations is outside the scope of State-Noisy MDP we analyzed in the last section and can have different impacts on the convergence of expectation-based and distributional RL algorithms. In this section, we firstly point out the vulnerability nature of expectation-based RL although we consider the bounded rewards in Section 4.1. Next, we characterize the robustness *blessing* of distributional RL based on Histogram distributional loss (Imani & White, 2018) with derived bounded gradient norms unlike the unrestricted one in expectation-based RL. Finally, as an extension we analyze the impact of more complex state observations on TD convergence and further conduct a sensitivity analysis by the influence function.

**Notation.** We derive theoretical results with the linear function approximator. For the expectation-based RL, the value estimate $\hat{v} : \mathcal{S} \times \mathbb{R}^d \to \mathbb{R}$ is formed simply as the inner product between state features $\mathbf{x}(s)$ and weights $\mathbf{w} \in \mathbb{R}^d$, given by $\hat{v}(s, \mathbf{w}) \overset{\text{def}}{=} \mathbf{w}^\top \mathbf{x}(s)$. At each step, the state feature can be rewritten as $\mathbf{x}_t \overset{\text{def}}{=} \mathbf{x}(S_t) \in \mathbb{R}^d$. The distributional RL setting is given in Section 4.2.

### 4.1 VULNERABILITY FOR EXPECTATION-BASED RL

In the classical RL with function approximation, a natural objective function is *Mean Squared Value Error* (Sutton & Barto, 2018) denoted $\overline{\text{VE}}$, which is weighted by the state distribution $\mu$:

$$\overline{\text{VE}}(\mathbf{w}) \doteq \sum_{s \in \mathcal{S}} \mu(s) \left[ v_\pi(s) - \hat{v}(s, \mathbf{w}) \right]^2, \tag{9}$$

where $\mu(s)$ is the fraction of time spent in $s$. To solve this kind of *weighted* least squared minimization, we leverage Stochastic Gradient Descent (SGD) and the weight updating rule can be formulated as

$$\begin{aligned} \mathbf{w}_{t+1} &\doteq \mathbf{w}_t + \alpha \left[ v_\pi(s_t) - \hat{v}(s_t, \mathbf{w}_t) \right] \nabla \hat{v}(s_t, \mathbf{w}_t) \\ &\doteq \mathbf{w}_t + \alpha \left( U_t - \mathbf{w}_t^\top \mathbf{x}_t \right) \mathbf{x}_t \end{aligned} \tag{10}$$

where $\alpha$ is the step size and the target $U_t$ can be either an unbiased estimate via Monte Carlo method with $U_t = \sum_{k=0}^\infty \gamma^k r_{t+k+1}$, or a biased estimate via TD learning with $U_t = r_{t+1} + \gamma \mathbf{w}_t^\top \mathbf{x}_{t+1}$. Let $g_{\overline{\text{VE}}}$ be the empirical version of $\overline{\text{VE}}(\mathbf{w})$. To investigate the vulnerability of expectation-based RL over state features, we focus on the gradient norm of $g_{\overline{\text{VE}}}$ regarding $\mathbf{x}(s)$ (or $\mathbf{x}_t$). For a fair comparison with distributional RL in Section 4.2, we additionally bound the norm of $\mathbf{w}$, i.e., $\|\mathbf{w}\| \leq l$, which can also be easily satisfied by imposing $\ell_1$ or $\ell_2$ regularization. Therefore, the upper bound of gradient norm of $g_{\overline{\text{VE}}}$ is

$$\| \frac{\partial}{\partial \mathbf{x}(s)} g_{\overline{\text{VE}}} \| = |U_t - \mathbf{w}_t^\top \mathbf{x}_t| \|\mathbf{w}_t\| \leq |U_t - \mathbf{w}_t^\top \mathbf{x}_t| l. \tag{11}$$

However, this upper bound $|U_t - \mathbf{w}_t^\top \mathbf{x}_t| l$ can be arbitrary large as there is no restriction on $|U_t - \mathbf{w}_t^\top \mathbf{x}_t|$ in either Monte Carlo or TD bootstrap estimate of $U_t$, which we elaborate as follows.

**Case 1: Estimate $U_t$ via Monte Carlo.** If we consider the bounded rewards, i.e., $r \in [R_{\min}, R_{\max}]$, we can bound $U_t$ as $U_t = \sum_{k=0}^\infty \gamma^k r_{t+k+1} \in [\frac{R_{\min}}{1-\gamma}, \frac{R_{\max}}{1-\gamma}]$. However, the bounded rewards can not guarantee the bounded gradient norm of $g_{\overline{\text{VE}}}$ as $\mathbf{w}_t^\top \mathbf{x}_t$ can be sufficiently large.

**Case 2: Estimate $U_t$ via bootstrap estimate, e.g., TD (0).** Consider TD (0) where $U_t = r_{t+1} + \gamma \mathbf{w}_t^\top \mathbf{x}_{t+1}$. However, we can find that $|U_t - \mathbf{w}_t^\top \mathbf{x}_t| = |r_{t+1} + \mathbf{w}_t^\top (\gamma \mathbf{x}_{t+1} - \mathbf{x}_t)|$ can still be sufficiently large due to the unrestricted term $\mathbf{w}_t^\top (\gamma \mathbf{x}_{t+1} - \mathbf{x}_t)$.

Therefore, we conclude that the unbounded gradient norm of objective function regarding state features for the expectation-based RL results in its vulnerability against state perturbations.

## 4.2 Robustness Blessing for distributional RL

We show that in the function approximation setting, the distributional loss in distributional RL can additionally yield *Lipschitz continuity* regarding state features, thus leading to more stable gradients relatively to expectation-based RL.

In particular, in distributional RL our goal is to minimize $\mathcal{L}(Z_\theta, \mathfrak{T}Z_\theta)$, where $\mathfrak{T}$ is the distributional Bellman operator. Here we leverage histogram to parameterize the distribution $Z_\theta$ based on KL divergence as $\mathcal{L}$, yielding the *histogram distributional loss* (Imani & White, 2018). Specifically, we uniformly partition the support of $\mathbf{x}(s)$ into $k$ bins, and let function $f : \mathcal{X} \rightarrow [0, 1]^k$ provide k-dimensional vector $f(\mathbf{x}(s))$ of the coefficients indicating the probability the target is in that bin given $\mathbf{x}(s)$. Next, we use *softmax* based on the linear approximation $\mathbf{x}(s)^\top \theta_i$ to express $f$, i.e., $f_i(\mathbf{x}(s)) = \exp\left(\mathbf{x}(s)^\top \theta_i\right) / \sum_{j=1}^k \exp\left(\mathbf{x}(s)^\top \theta_j\right)$. Therefore, the histogram distributional loss $\mathcal{L}(Z_\theta, \mathfrak{T}Z_\theta)$ between $Z_\theta$ and $\mathfrak{T}Z_\theta$ can be derived as

$$\mathcal{L}_\theta = -\sum_{i=1}^k p_i \log f_i^\theta(\mathbf{x}(s)), \tag{12}$$

where $\theta = \{\theta_1, ..., \theta_k\}$ and the target probability $p_i$ is the cumulative probability increment of target distribution $\mathfrak{T}Z_\theta$ within the $i$-th bin. Detailed derivation of the histogram distributional loss is given in Appendix C.

Based on this histogram distributional loss in distribution RL, we obtain Theorem 2 (proof in Appendix C), revealing that the distribution loss can result in additional Lipschitz continuity property that bounds the norm of gradient over state features $\mathbf{x}(s)$:

**Theorem 2.** *(Lipschitz Continuity of distributional RL) Consider the histogram distributional loss* $\mathcal{L}_\theta = -\sum_{i=1}^k p_i \log f_i^\theta(\mathbf{x}(s))$, *where* $f_i(\mathbf{x}(s)) = \exp\left(\mathbf{x}(s)^\top \theta_i\right) / \sum_{j=1}^k \exp\left(\mathbf{x}(s)^\top \theta_j\right)$ *parameterized by* $\theta = \{\theta_1, ..., \theta_k\}$. *Assume* $\|\theta_i\| \leq l$ *for* $\forall i = 1, .., k$, *then* $\mathcal{L}_\theta$ *is kl-Lipschitz continuous w.r.t.* $\mathbf{x}(s)$, *yielding a bounded norm of its gradient, i.e.,*

$$\left\| \frac{\partial}{\partial \mathbf{x}(s)} \mathcal{L}_\theta \right\| \leq kl \tag{13}$$

In conclusion, Theorem 2 shows that distributional loss in distributional RL can additionally enjoy $kl$-Lipschitz continuity compared with the expectation-based RL where the gradient norm of objective function is unbounded. Thus, the bounded gradient norm regarding state features of distributional RL mitigates the impact of state noises on the objective function while training, therefore yielding better training robustness.

**Extension of TD Convergence and Sensitivity Analysis.** Beyond the aforementioned robustness analysis on both expectation-based and distributional RL, we also conduct the analysis on the perturbation impacts of different state observations in the TD learning based on *expectation-based RL*, including the current, next and both state observations. In particular, conditions for TD convergence in these three different noisy state observation settings are derived in Theorem 4 of Appendix D, which are strongly related to the positive definiteness property of relevant matrixs. More importantly, the convergence conditions for current and next noisy state observations are harder than the condition in the noise-free setting and are normally divergent. Nevertheless, which situation is milder heavily depends on the task. Interested readers can refer to Appendix D for more details. In addition, we also conduct a sensitivity analysis by the *influence function* to characterize the effects that the state noise in particular observation has on an estimator in Theorem 5 of Appendix E. The conclusions we made are similar to the TD convergence analysis part. Note that although the extension analysis is based on the *expection-based RL with linear function approximation*, our empirical observations demonstrate our analysis results *across both expectation-based and distributional RL*. We defer detailed empirical verification in the following experimental section.

**Remark.** In this section, we firstly present the vulnerability of expectation-based RL against noisy state observations in Section 4.1, while we further derived the Lipschitz continuity blessing of distributional RL in Section 4.2, leading to more robustness against noisy state observations than expectation-based RL. In Section 5, we show that our empirical observations coincide with our theoretical results to further demonstrate the robustness advantage of distributional RL.

## 5 EXPERIMENTS

We make a comparison between expectation-based and distributional RL algorithms against various noisy state observations. We select DQN (Mnih et al., 2015) as the baseline, and QR-DQN (Dabney et al., 2018b) as its distributional counterpart. The previous analysis is either for policy evaluation or linear function approximation, but there are natural—though in some cases heuristic—extensions to the control setting and to non-linear function approximation.

**Experimental Setup.** We perform our algorithms on Cart Pole, Mountain Car, Breakout and Qbert games. We followed the procedure in (Ghiassian et al., 2020; Zhang & Yao, 2019). All the experimental settings, including parameters, are identical to the distributional RL baselines implemented by (Zhang, 2018; Dabney et al., 2018b). We perform 200 runs on both Cart Pole and Mountain Car and 3 runs on Breakout and Qbert. Reported Results are averaged with shading indicating the standard error. The learning curve is smoothed over a window of size 10 before averaging across runs. Please refer to Appendix H for more details.

**Noisy State Observations.** For the random noise, we use Gaussian noise with different standard deviations. In particular, for a better presentation to compare the difference, we select the standard deviations as 0.05, 0.1 in Cart Pole, 0.01, 0.0125 in Mountain Car, 0.01, 0.05 in Breakout and 0.05 in Qbert. For the adversarial noise, we followed (Zhang et al., 2020), where the set of noises $B(s)$ is defined as an $\ell_\infty$ norm ball around $s$ with a radius $\epsilon$, given by $\ell_\infty B(s) := \{\hat{s} : \|s - \hat{s}\|_\infty \leq \epsilon\}$. We apply Projected Gradient Descent (PGD) version in (Pattanaik et al., 2017), with 3 fixed iterations while adjusting $\epsilon$ to control the perturbation strength. Concretely, for better presentation we select the perturbation size $\epsilon$ as 0.05, 0.1 in Cart Pole, 0.01, 0.1 in Mountain Car, 0.005, 0.01 in Breakout, and 0.005 in Qbert.

**Experiments Structure.** In Section 5.1, we consider the function approximation setting with TD learning where both current and next state observations are perturbed, leading to a SN-MDP-like setting as the vanilla SN-MDP is a tabular case. In this case, we found both DQN and QRDQN converge to different convergence points determined by the noise strength. Next, in Section 5.2 and 5.3, we investigate the robustness advantage of distributional RL, i.e., QRDQN, relative to expectation-based RL, i.e., DQN, under various random and adversarial state observations, respectively.

### 5.1 SN-MDP-LIKE SETTING

In the classical SN-MDP tabular case, each state will be perturbed via the noise mechanism $N$. We keep this noise structure and transform it into the function approximation setting with TD learning,

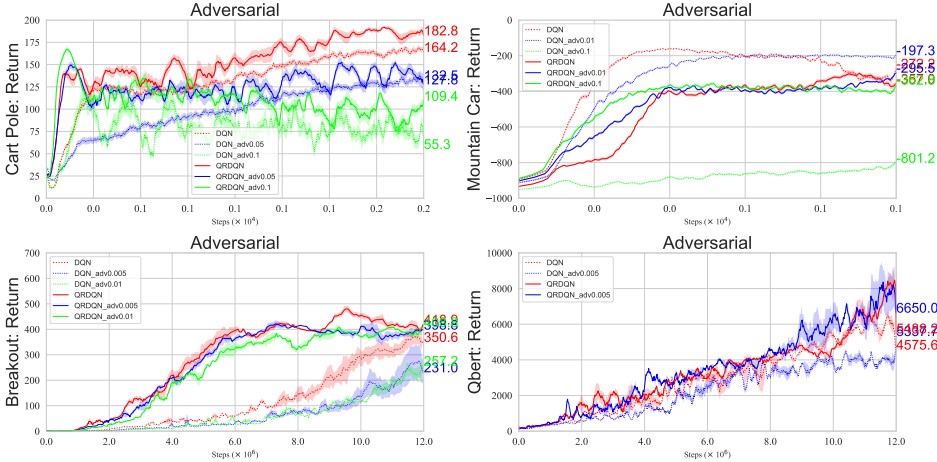

Figure 2: Average returns of DQN and QRDQN against **adversarial** state observation noises across four games. **advX** in the legend indicates adversarial state observations with the perturbation size $\epsilon$ as **X**. Both QRDQN (solid lines) and DQN (dashed lines) converge and the convergence level is determined by the perturbation strength $\epsilon$.

and thus both current and next state observations would be noisy. We further investigate the training robustness of both DQN and QRDQN against either random or adversarial state perturbations across these four games.

We focus on the adversarial setting as suggested in Figure 2. It turns out that all DQN and QRDQN algorithms under various strength of adversarial state noises converge, although they eventually obtained different average returns. This empirical observation is consistent with our theoretical analysis in Section 3 where both classical and distributional Bellman operators are contractive and thus convergent. In addition, Figure 5 also manifests that the final performance that algorithms attained has a decreasing tendency as the perturbation strength, i.e., standard deviation, increases especially for DQN. Remarkably, the final performance of QRDQN is more robust than DQN against different perturbation strength, especially in Breakout and Qbert games, although both DQN and QRDQN are convergent in this SN-MDP-like setting. For the random state observation setting, we provide the similar results in Figure 5 with detailed explanation in Appendix I.

## 5.2 Setting Beyond SN-MDP with Random State Noises

Beyond SN-MDP, we further investigate the training robustness of DQN and QRDQN in the more complicated unbalanced state observation setting as mentioned in Section 4, where the agent observe perturbations on *current* state observations. Similar results in the setting with the perturbed *next* state observations are given in Appendix J. In this subsection, we firstly focus on the function approximation case with random noises on current state observations, and the learning curves when the agent encounters different strength of random state noises are provided in Figure 3 across four games.

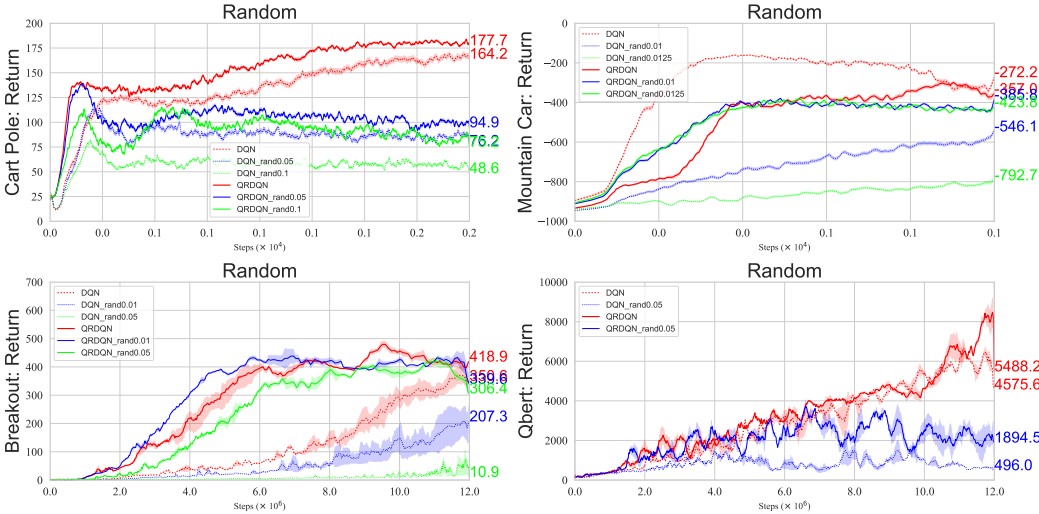

Figure 3: Average returns of DQN and QRDQN against **random** state observation noises across four games. **randX** in the legend indicates random state observations with the standard deviation **X**. QRDQN (solid lines) almost consistently outperform DQN (dashed lines) given the same perturbation strength, yielding more training robustness against noisy state observations.

From Figure 3, it illustrates that the final performance of QRDQN (solid lines) is superior to DQN (DQN) in the same color given the same random state perturbations (the same standard deviation) when both DQN and QRDQN achieved similar average returns eventually (red dashed and solid lines). This implies that QRDQN enjoys better training robustness than DQN under the random noisy state observation setting. It is worthy to note that in Mountain Car game although DQN outperforms QRDQN almost across all training process, QRDQN under random state noises (blue and green lines) can surpass DQN with the same noise setting, yielding more training robustness. A more significant result is in Breakout where the training of QRDQN is less vulnerable to various random state perturbations, while the performance of DQN degrades and even diverges as the perturbation strength increases.

## 5.3 SETTING BEYOND SN-MDP WITH ADVERSARIAL STATE NOISES

Next, we probe the training robust of DQN and QRDQN in the setting where the agent encounters the *adversarial* state observations in the current state in the function approximation case. Figure 4 presents the learning curves of algorithms over four games against noisy states under different adversarial perturbation sizes $\epsilon$.

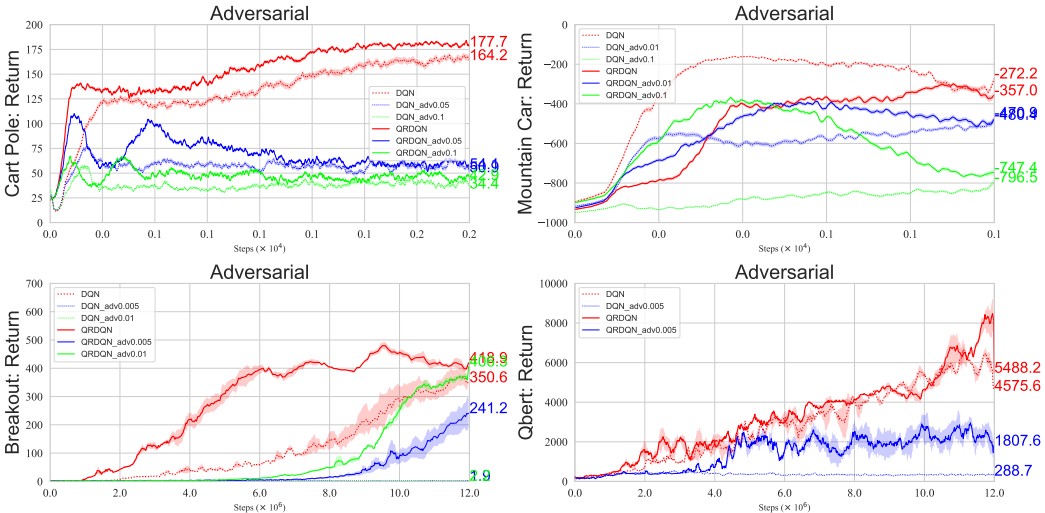

Figure 4: Average returns of DQN and QRDQN against **adversarial** state observation noises across four games. **advX** in the legend indicates random state observations with the perturbation size $\epsilon$ **X**. QRDQN (solid lines) almost consistently outperform DQN (dashed lines) given the same perturbation strength, yielding more training robustness against noisy state observations.

The results under the adversarial state observations are similar to those in the random noises case. Specifically, both DQN and QRDQN algorithms tend to degrade when getting exposed to adversarial state observations, and even are more likely to diverge. For example, in Cart Pole both DQN (blue and green dotted lines) and QRDQN (blue and green solid lines) degraded to similar poor results when exposing to strong perturbation with $\epsilon = 0.05$ and $0.1$. However, a key observation is that *QRDQN is capable of obtaining desirable performance while DQN even diverges*. For instance, in both Breakout and Qbert game, DQN (dotted blue lines) under the adversarial perturbation with $\epsilon = 0.005$ led to divergence, QRDQN (solid blue lines) can still attain relatively satisfactory performance eventually on the condition that the performance of both two algorithms are on part with each other without any noisy state observations (red lines).

**Sensitivity Analysis of Different Perturbed States.** To demonstrate the theoretical results (Theorem 4 and 5) in the Appendix based on TD convergence condition and influence function we ever mentioned in Section 4.2, we provide the consistent results in Appendix D and E. Interested readers can refer to Appendix D and E for a more detailed discussion.

## 6 DISCUSSION AND CONCLUSION

The Lipschitz continuity blessing is based on the histogram distributional loss, but it is more expected that similar conclusions can be made under Wasserstein or Crammer distance as these distances are more approachable in real distributional RL algorithms. We leave it as future works.

In this paper, we explored the training robustness of distributional RL against noisy state observations. After the convergence analysis of distributional RL in the SN-MDP, we proved the Lipschitz continuity property of distributional RL, accounting for its less vulnerability. We also provided the TD convergence conditions and a sensitivity analysis on more complex noisy settings. Experimental observations coincides with our theoretical results.

**Ethics Statement.** Our works reveals that distributional RL can enjoy the training robustness against noisy state observations. The advantage is useful to defend against the poisoning attacks, thus contributing to the privacy of algorithms. Based on our experience, there is no other ethic concerns of our work.

**Reproducibility Statement.** For the theoretical part, we clearly state the related assumption and detailed proof process in the appendix. In terms of the algorithm, our implementation is directly adapted from the public RL algorithms, including DQN and QR-DQN.

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

## A  THEOREM 3 WITH PROOF

**Theorem 3.** *(Convergence and Contraction of Bellman operators in the SN-MDP) Given a policy $\pi$, define the Bellman operator $\mathcal{T} : \mathbb{R}^{|S|} \to \mathbb{R}^{|S|}$ under random and adversarial states noises by $\mathcal{T}_r^\pi$ and $\mathcal{T}_a^\pi$, respectively. Denote a "merged" policy $\pi'$ where $\pi'(a|s) = \sum_{v(s)} N(v(s)|s)\pi(a|v(s))$ and $\mathbf{S}(\pi)$ is a policy set given $\pi$. Then we have:*

*(1) $\mathcal{T}_r^\pi$ is a contraction operator and can converge to $V_{\pi'}$, i.e., $\mathcal{T}_r^\pi \tilde{V}_{\pi \circ N} = \tilde{V}_{\pi \circ N} = V_{\pi'}$, where multiple policies $\pi_r \in \mathbf{S}(\pi)$ might exist with $\sum_{v(s)} N(v(s)|s)\pi_r(a|v(s)) = \pi'(a|s)$.*

*(2) $\mathcal{T}_a^\pi$ is a contraction with the convergence satisfying $\mathcal{T}_a^\pi \tilde{V}_{\pi \circ N^*} = \min_N \tilde{V}_{\pi \circ N} = V_{\pi \circ N^*}$, where $N^*$ is the optimal adversarial noise strategy. If the optimal policy $\pi_a$ exists, it satisfies $\pi_a(a|v^*(s)) = \pi(a|s)$ for each $s$ and $a$, where $v^*(s)$ is the adversarial noisy state manipulated by $N^*(\cdot|s)$.*

*Proof.* Our proof is partly based on Theorem 1 and 2 in (Zhang et al., 2020), but adds more analysis on the converged policy especially under the random noisy states setting. The most important insight in the following proof is that the noise transition can be merged into the agent's policy, resulting in a new "merged" policy $\pi'$.

**Proof of (1)**  Firstly, as the Bellman Equation under the random noisy states is right the general form in Eq. 6, it automatically satisfies that $\mathcal{T}_r^\pi \tilde{V}_{\pi \circ N} = \tilde{V}_{\pi \circ N}$ when it converges. As for the proof of contraction, based on our insight about the new "merged" policy $\pi'$ where $\pi'(a|s) = \sum_{v(s)} N(v(s)|s)\pi(a|v(s))$, we can rewrite our Bellman Operator as:

$$
\begin{aligned}
&\mathcal{T}_r^\pi \tilde{V}_{\pi \circ N}(s) \\
&= \sum_a \pi'(a|s) \sum_{s'} p(s'|s,a) \left[ R(s,a,s') + \gamma \tilde{V}_{\pi \circ N}(s') \right] \\
&= \mathbf{R}(s) + \gamma \sum_{s'} \mathbf{P}'_{s,s'} \tilde{V}_{\pi \circ N}(s')
\end{aligned}
\tag{14}
$$

where $\mathbf{R}(s) = \sum_a \pi'(a|s) \sum_{s'} p(s'|s,a) R(s,a,s')$, and $\mathbf{P}'_{s,s'} = \sum_a \pi'(a|s) p(s'|s,a)$ determined by the "merged" policy $\pi'$. Then for two different value function $\tilde{V}_{\pi \circ N}^1$ and $\tilde{V}_{\pi \circ N}^2$ we have:

$$
\begin{aligned}
&\|\mathcal{T}_r^\pi \tilde{V}_{\pi \circ N}^1 - \mathcal{T}_r^\pi \tilde{V}_{\pi \circ N}^2\|_\infty \\
&= \max_s |\gamma \sum_{s'} \mathbf{P}'_{s,s'} \tilde{V}_{\pi \circ N}^1(s') - \gamma \sum_{s'} \mathbf{P}'_{s,s'} \tilde{V}_{\pi \circ N}^2(s')| \\
&\leq \gamma \max_s \sum_{s'} \mathbf{P}'_{s,s'} |\tilde{V}_{\pi \circ N}^1(s') - \tilde{V}_{\pi \circ N}^2(s')| \\
&\leq \gamma \max_s \sum_{s'} \mathbf{P}'_{s,s'} \max_{s'} |\tilde{V}_{\pi \circ N}^1(s') - \tilde{V}_{\pi \circ N}^2(s')| \\
&= \gamma \max_s \sum_{s'} \mathbf{P}'_{s,s'} \|\tilde{V}_{\pi \circ N}^1 - \tilde{V}_{\pi \circ N}^2\|_\infty \\
&= \gamma \|\tilde{V}_{\pi \circ N}^1 - \tilde{V}_{\pi \circ N}^2\|_\infty
\end{aligned}
\tag{15}
$$

Then according to the Banach fixed-point theorem, since $\gamma \in (0,1)$, $\tilde{V}_{\pi \circ N}$ converges to a unique fixed-point $V_{\pi'}$. However, even though the obtained policy $\pi'$ satisfies that $\pi'(a|s) = \sum_{v(s)} N(v(s)|s)\pi(a|v(s))$ for each $s, a$, these equations can not necessarily guarantee a unique $\pi$ especially when these equations behind this condition are underdetermined. In such scenario, multiple policies $\pi_r$ will exist as long as they satisfy the equations above.

**Proof of (2)**  Firstly, based on Theorem 1 (Zhang et al., 2020) that shows an optimal policy does not always exist, we assume that an optimal policy exists in the adversarial noisy state setting for the convenience of following analysis. Based on this assumption, we need to derive the explicit value function under the adversary. Inspired by (Zhang et al., 2020), the proof insight is that the behavior of optimal adversary can be also viewed as finding another optimal policy, yielding a zero-sum two player game. Specifically, in the SN-MDP setting, the adversary selects an action $\hat{a} \in \mathcal{S}$ satisfying $\hat{a} = v(s)$, attempting to maximize its state-action value function

$\tilde{Q}_{\pi_a}(s, \hat{a})$. Then the adversary's value function $\hat{V}_{\pi_a}(s)$ can be formulated as:

$$
\begin{aligned}
\hat{V}_{\pi_a}(s) &= \max_{\hat{a}} \hat{Q}_{\pi_a}(s, \hat{a}) \\
&= \max_{\hat{a}} \sum_{s'} \hat{p}(s'|s, \hat{a})(\hat{R}(s, \hat{a}, s') + \gamma \hat{V}_{\pi_a}(s')) \\
&= \max_{v(s)} \sum_{s'} \sum_{a} \pi(a|v(s))p(s'|s, a)(-R(s, a, s') \\
&\quad + \gamma \hat{V}_{\pi_a}(s'))
\end{aligned}
\tag{16}
$$

where $\hat{p}(s'|s, \hat{a})$ is the transition dynamics of the adversary, satisfying $\hat{p}(s'|s, \hat{a}) = \sum_a \pi(a|v(s))p(s'|s, a)$ from the perspective of the agent. $\hat{R}(s, \hat{a}, s')$ is the adversary's reward function while taking action $\hat{a}$, which is the opposite number of $R(s, a, s')$ given the action $a$. In addition, since both the adversary and agent can serve as a zero-sum two-player game, it indicates that $\tilde{V}_{\pi_a}(s) = -\hat{V}_{\pi_a}(s)$ for the agent's value function $\tilde{V}_{\pi_a}$ in the adversary setting. Then we rearrange the equation above as follows:

$$
\begin{aligned}
\tilde{V}_{\pi_a}(s) &= -\hat{V}_{\pi_a}(s) \\
&= -\min_{N(\cdot|s)} \sum_{s'} \sum_{a} \pi'(a|s)p(s'|s, a)(-R(s, a, s') \\
&\quad - \gamma \tilde{V}_{\pi_a}(s')) \\
&= \min_{v(s)} \sum_{s'} \sum_{a} \pi'(a|s)p(s'|s, a)(R(s, a, s') \\
&\quad + \gamma \tilde{V}_{\pi_a}(s')) \\
&= \min_{N(\cdot|s)} \sum_{s'} \sum_{a} \pi'(a|s)p(s'|s, a)(r_{t+1} + \gamma \min_{N} \mathbb{E}_{\pi \circ N} \left[ \sum_{k=0}^{\infty} r_{t+k+2} | s_{t+1} = s' \right]) \\
&= \min_{N} \tilde{V}_{\pi \circ N}(s)
\end{aligned}
\tag{17}
$$

Note that we optimize over $N$, which means we consider $N(\cdot|s)$ for each state $s$. Further, we derive the contraction of the Bellman operator $\mathcal{T}_a^\pi$. We rewrite our Bellman Operator $\mathcal{T}_a^\pi$ as:

$$
\begin{aligned}
\mathcal{T}_a^\pi \tilde{V}_{\pi \circ N}(s) &= \min_{N} \tilde{V}_{\pi \circ N}(s) \\
&= \min_{N} \boldsymbol{R}(s) + \gamma \sum_{s'} \boldsymbol{P}'_{s, s'} \tilde{V}_{\pi \circ N}(s')
\end{aligned}
\tag{18}
$$

We firstly assume $\mathcal{T}_a^\pi \tilde{V}_{\pi_a}^1(s) \geq \mathcal{T}_a^\pi \tilde{V}_{\pi_a}^2(s)$, then we have:

$$
\begin{aligned}
&\mathcal{T}_a^\pi \tilde{V}_{\pi \circ N}^1(s) - \mathcal{T}_a^\pi \tilde{V}_{\pi \circ N}^2(s) \\
&\leq \max_{N(\cdot|s)} \{ \gamma \sum_{s'} \boldsymbol{P}'_{s, s'} \tilde{V}_{\pi \circ N}^1(s') - \gamma \sum_{s'} \boldsymbol{P}'_{s, s'} \tilde{V}_{\pi \circ N}^2(s') \} \\
&\leq \gamma \max_{N(\cdot|s)} \sum_{s'} \boldsymbol{P}'_{s, s'} | \tilde{V}_{\pi \circ N}^1(s') - \tilde{V}_{\pi \circ N}^2(s') | \\
&\leq \gamma \max_{N(\cdot|s)} \sum_{s'} \boldsymbol{P}'_{s, s'} \max_{s} | \tilde{V}_{\pi \circ N}^1(s') - \tilde{V}_{\pi \circ N}^2(s') | \\
&= \gamma \max_{N(\cdot|s)} \sum_{s'} \boldsymbol{P}'_{s, s'} \| \tilde{V}_{\pi \circ N}^1 - \tilde{V}_{\pi \circ N}^2 \|_{\infty} \\
&\leq \gamma \| \tilde{V}_{\pi \circ N}^1 - \tilde{V}_{\pi \circ N}^2 \|_{\infty}
\end{aligned}
\tag{19}
$$

where the first inequality holds as $\min_{x_1} f(x_1) - \min_{x_2} g(x_2) \leq \max_x (f(x) - g(x))$ and we extends this inequality into the Wasserstein distance in the proof of convergence of distributional RL setting in Appendix B. The last inequality holds since only $\boldsymbol{P}'_{s, s'}$ depends on $N(\cdot|s)$ while the infinity norm is a constant, which is independent with the current $N(\cdot|s)$. Similarly, the other scenario can be still proved. Thus, we have:

$$
\| \mathcal{T}_a^\pi \tilde{V}_{\pi \circ N}^1 - \mathcal{T}_a^\pi \tilde{V}_{\pi \circ N}^2 \|_{\infty} \leq \gamma \| \tilde{V}_{\pi \circ N}^1 - \tilde{V}_{\pi \circ N}^2 \|_{\infty}
\tag{20}
$$

Thus, we proved that $\mathcal{T}_a^\pi$ is still a contraction and converge to $\min_N \tilde{V}_{\pi \circ N}$. We denote it as $\tilde{V}_{\pi \circ N^*}$ In addition, based on the insight of the "merged" policy $\pi_a'$, we have $\pi_a' = \sum_{v(s)} N^*(v(s)|s)\pi(a|v(s)) = \pi(a|v^*(s))$ where the deterministic state $v^*(s)$ is the adversarial noisy state from the state $s$.

$\square$

# B  PROOF OF THEOREM 1

*Proof.* Firstly, we will provide the properties of Wassertein distance $d_p$ in Lemma 1 that we leverage in our following convergence proof.

**Lemma 1.** *(Properties of Wasserstein Metric) We consider the distribution distance between the random variable $U$ and $V$. Denote $d_p$ as the Wasserstein distance between two distribution defined in Eq. 4. For any scalar $a$ and random variable $A$ independent of $U$ and $V$, the following relationships hold:*

$$d_p(aU, aV) \leq |a| d_p(U, V)$$
$$d_p(A + U, A + V) \leq d_p(U, V) \tag{21}$$
$$d_p(AU, AV) \leq \|A\|_p d_p(U, V)$$

*Further, let $A_1, A_2, ...$ be a set of random variables describing the a partition of $\omega$, when the partition lemma holds:*

$$d_p(U, V) \leq \sum_i d_p(A_i U, A_i V). \tag{22}$$

Then, the following contraction proof is in the maximal form of $d_p$ and we denote it as $\bar{d}_p$.

**Proof of (1)**   This contraction proof is similar to the original one (Bellemare et al., 2017) in the distributional RL without state observation noises. The only difference lies in the new transition operator $\mathcal{P}_r^\pi$, but it dose not change the main proof process. For two different random variables $Z_N^1$ and $Z_N^2$ about returns, we have:

$$
\begin{aligned}
&\bar{d}_p(\mathfrak{T}_r^\pi Z_N^1, \mathfrak{T}_r^\pi Z_N^2) \\
&= \sup_{s,a} d_p(\mathfrak{T}_r^\pi Z_N^1(s, a), \mathfrak{T}_r^\pi Z_N^2(s, a)) \\
&= \sup_{s,a} d_p(R(s, a, S') + \gamma \mathcal{P}_r^\pi Z_N^1(s, a), R(s, a, S') + \gamma \mathcal{P}_r^\pi Z_N^2(s, a)) \\
&\leq \gamma \sup_{s,a} d_p(\mathcal{P}_r^\pi Z_N^1(s, a), \mathcal{P}_r^\pi Z_N^2(s, a)) \\
&\leq \gamma \sup_{s,a} \sup_{s',a'} d_p(Z_N^1(s', a'), Z_N^2(s', a')) \\
&= \gamma \sup_{s',a'} d_p(Z_1(s', a'), Z_2(s', a')) \\
&= \gamma \sup_{s,a} d_p(Z_N^1(s, a), Z_N^2(s, a)) \\
&= \gamma \bar{d}_p(Z_N^1, Z_N^2).
\end{aligned}
\tag{23}
$$

Thus, we conclude that $\mathfrak{T}_r^\pi : \mathcal{Z} \to \mathcal{Z}$ is a $\gamma$-contraction in $\bar{d}_p$.

**Proof of (2)**   Firstly, we define the distributional Bellman optimality operator $\mathfrak{T}$ in MDP as

$$\mathfrak{T}Z(s, a) :\stackrel{D}{=} R\left(s, a, S'\right) + \gamma Z(S', \pi_Z(s')) \tag{24}$$

where $S' \sim P(\cdot|s, a)$ and $\pi_Z(S') = \arg\max_{a'} \mathbb{E}[Z(S', a')]$. By contrast, in SN-MDP, Our greedy adversarial rule $N^*(\cdot|s')$ is based on the greedy policy rule in distributional Bellman optimality operator, which attempts to find adversarial $N^*(\cdot|s')$ in order to minimize $\mathbb{E}[Z_N(s', a')]$, where $a' \sim \pi(\cdot|V(s'))$ and $V(s') \sim N(\cdot|s')$. We assume $N^*(\cdot|s')$ yields a deterministic state $s^*$, and thus the agent always takes action based on $s^*$, which we denote as $A^* \sim \pi(\cdot|s^*)$. Therefore, we can obtain the state-action function $Q_{N^*}^\pi(s, a)$ under the adversary as

$$
\begin{aligned}
Q_{N^*}^\pi(s, a) &= \min_N \mathbb{E}[Z_N^\pi(s, a)] \\
&= \mathbb{E}\left[Z^{\pi^*}(s, a)\right]
\end{aligned}
\tag{25}
$$

where $\pi^*(\cdot|s) = \pi(\cdot|s^*)$ for $\forall s$ that follows the adversarial policy $A^*$.

Next, to derive the contractive property of $\mathfrak{T}_a^\pi$, we denote two state-action valued distributions as $Z_N^1(s,a)$ and $Z_N^2(s,a)$. Then we have:

$$
\begin{aligned}
&\bar{d}_p(\mathfrak{T}_a^\pi Z_N^1, \mathfrak{T}_a^\pi Z_N^2) \\
&= \sup_{s,a} d_p(\mathfrak{T}_a^\pi Z_N^1(s,a), \mathfrak{T}_a^\pi Z_N^2(s,a)) \\
&= \sup_{s,a} d_p(R(s,a,S') + \gamma \mathcal{P}_a^\pi Z_N^1(s,a), R(s,a,S') + \gamma \mathcal{P}_a^\pi Z_N^2(s,a)) \\
&\leq \gamma \sup_{s,a} \sum_{s'} P(s'|s,a) d_p(Z_N^1(s',A^*), Z_N^2(s',A^*)) \\
&= \gamma \sum_{s'} P(s'|s,a) d_p(Z_N^1(s',A^*), Z_N^2(s',A^*)) \\
&\leq \gamma \sup_{s'} d_p(Z_N^1(s',A^*), Z_N^2(s',A^*)) \\
&= \gamma \sup_{s'} d_p\Big(\sum_{a_*'} \pi(a_*'|s^*) Z_N^1(s',a_*'), \sum_{a_*'} \pi(a_*'|s^*) Z_N^2(s',a_*')\Big) \\
&\leq \gamma \sup_{s'} \sum_{a_*'} \pi(a_*'|s^*) d_p(Z_N^1(s',a_*'), Z_N^2(s',a_*')) \\
&\leq \gamma \sup_{s',a_*'} d_p(Z_N^1(s',a_*'), Z_N^2(s',a_*')) \\
&= \gamma \sup_{s,a} d_p(Z_N^1(s,a), Z_N^2(s,a)) \\
&= \bar{d}_p(Z_N^1, Z_N^2)
\end{aligned}
\tag{26}
$$

Thus, we conclude that $\mathfrak{T}_a^\pi$ is still a $\gamma$-contraction in $\bar{d}_p$. $\qquad\square$

## C  PROOF OF THEOREM 2

*Proof.* Firstly, we show the derivation details of the Histogram distribution loss starting from KL divergence between $p$ and $q_\theta$. $p_i$ is the cumulative probability increment of target distribution $\mathfrak{T}Z_\theta$ within the $i$-th bin, and $q_\theta$ corresponds to a (normalized) histogram, and has density values $\frac{f_i^\theta(\mathbf{x}(s))}{w_i}$ per bin. Thus, we have:

$$
\begin{aligned}
\mathcal{L}(Z_\theta, \mathfrak{T}Z_\theta) &= -\int_a^b p(y) \log q_\theta(y) dy \\
&= -\sum_{i=1}^k \int_{l_i}^{l_i+w_i} p(y) \log \frac{f_i^\theta(\mathbf{x}(s))}{w_i} dy \\
&= -\sum_{i=1}^k \log \frac{f_i^\theta(\mathbf{x}(s))}{w_i} \underbrace{(F_{\mathfrak{T}Z_\theta}(l_i+w_i) - F_{\mathfrak{T}Z_\theta}(l_i))}_{p_i} \\
&\doteq -\sum_{i=1}^k p_i \log f_i^\theta(\mathbf{x}(s))
\end{aligned}
\tag{27}
$$

where the last equality holds because the width parameter $w_i$ can be ignored for this minimization problem.

Next, we compute the gradient of the Histogram distributional loss.

$$
\frac{\partial}{\partial \mathbf{x}(s)} \sum_{j=1}^{k} p_j \log f_j^\theta(\mathbf{x}(s))
$$

$$
= \sum_{j=1}^{k} p_j \frac{1}{f_j^\theta(\mathbf{x}(s))} \nabla f_j^\theta(\mathbf{x}(s))
$$

$$
= \sum_{j=1}^{k} p_j \frac{1}{f_j^\theta(\mathbf{x}(s))} f_j^\theta(\mathbf{x}(s)) \sum_{i=1}^{k} \frac{\exp(\mathbf{x}(s)^\top \theta_i)}{\sum_{p=1}^{k} \exp(\mathbf{x}(s)^\top \theta_p)} (\theta_j - \theta_i)
$$

$$
= \sum_{j=1}^{k} p_j \sum_{i=1}^{k} f_i^\theta(\mathbf{x}(s))(\theta_j - \theta_i) \tag{28}
$$

$$
= \sum_{j=1}^{k} p_j \theta_j - \sum_{i=1}^{k} f_i^\theta(\mathbf{x}(s))\theta_i
$$

$$
= \sum_{i=1}^{k} (p_i - f_i^\theta(\mathbf{x}(s)))\theta_i
$$

Then, as we have $\|\theta_i\| \le l$ for $\forall i$, we bound the norm of its gradient

$$
\|\frac{\partial}{\partial \mathbf{x}(s)} \sum_{j=1}^{k} p_j \log f_j^\theta(\mathbf{x}(s))\|
$$

$$
\le \sum_{i=1}^{k} \|(p_i - f_i^\theta(\mathbf{x}(s)))\theta_i\| \tag{29}
$$

$$
= \sum_{i=1}^{k} |p_i - f_i^\theta(\mathbf{x}(s))| \|\theta_i\|
$$

$$
\le kl
$$

The last equality satisfies because $|p_i - f_i^\theta(\mathbf{x}(s))|$ is less than 1 and even smaller. In summary, compared with the least squared loss in expectation-based RL, the histogram distributional loss in distributional RL can additionally enjoy $kl$-Lipschitz continuity with bounded gradient norm regarding the state features $\mathbf{x}(s)$. This upper bound of gradient norm can mitigate the impact of the noises on state observations on the loss function, therefore yielding training robustness for distributional RL.

$\square$

# D  TD CONVERGENCE UNDER NOISY STATE OBSERVATIONS

Let $\mu(s)$ be the stationary distribution under the policy $\pi$ and $p(s'|s)$ be the transition probability from $s$ to $s'$ satisfying $p(s'|s) = \sum_a \pi(a|s)p(s'|s,a)$. We analyze conditions of TD convergence when exposing state observation noises. Firstly, we recall the classical TD update at step $t$:

$$
\mathbf{w}_{t+1} \leftarrow \mathbf{w}_t + \alpha_t(R_{t+1} + \gamma \mathbf{w}_t^\top \mathbf{x}_{t+1} - \mathbf{w}_t^\top \mathbf{x}_t)\mathbf{x}_t \tag{30}
$$

where $\alpha_t$ is the step size at time $t$. Once the system has reached the steady state for any $\mathbf{w}_t$, then the expected next weight vector can be written as $\mathbb{E}[\mathbf{w}_{t+1}|\mathbf{w}_t] = \mathbf{w}_t + \alpha_t(\mathbf{b} - \mathbf{A}\mathbf{w}_t)$, where $\mathbf{b} = \mathbb{E}(R_{t+1}\mathbf{x}_t) \in \mathbb{R}^d$ and $\mathbf{A} \doteq \mathbb{E}\left[\mathbf{x}_t d_t^\top\right] \in \mathbb{R}^{d \times d}$. The TD fixed point $\mathbf{w}_{\text{TD}}$ to the system satisfies $\mathbf{A}\mathbf{w}_{\text{TD}} = \mathbf{b}$. From (Sutton & Barto, 2018), we know that the matrix $\mathbf{A}$ determines the convergence in the linear TD setting. In particular, $\mathbf{w}_t$ converges with probability one to the TD fixed point if $\mathbf{A}$ is positive definite. However, if we add state noises $\eta$ on either $\mathbf{x}_t$ or $\mathbf{x}_{t+1}$ in Eq. 30, the convergence condition will be different. Theorem 4 (proof in Appendix F) provides conditions for TD convergence in three different noisy state observation settings.

**Theorem 4.** *(Conditions for TD Convergence under Noisy State Observations) Define $\mathbf{P}$ as the $|\mathcal{S}| \times |\mathcal{S}|$ matrix forming from $p(s'|s)$, $\mathbf{D}$ as the $|\mathcal{S}| \times |\mathcal{S}|$ diagonal matrix with $\mu(s)$ on its diagonal, and $\mathbf{X}$ as the $|\mathcal{S}| \times d$ matrix with $\mathbf{x}(s)$ as its rows, and $\mathbf{E}$ is the $|\mathcal{S}| \times d$ perturbation matrix with each perturbation vector $\mathbf{e}(s)$ as its rows. The stepsizes $\alpha_t \in (0,1]$ satisfy $\sum_{t=0}^{\infty} \alpha_t = \infty$ and $\sum_{t=0}^{\infty} \alpha_t^2 = 0$. For noisy states, we consider the following three cases: (i) $\mathbf{e}(s)$ on current state features, i.e., $\mathbf{x}_t \leftarrow \mathbf{x}_t + \mathbf{e}_t$, (ii) $\mathbf{e}(s')$ on next state features, i.e., $\mathbf{x}_{t+1} \leftarrow \mathbf{x}_{t+1} + \mathbf{e}_{t+1}$, (iii) the same $\mathbf{e}$ on both state features. We can attain that $\mathbf{w}_t$ converges to TD fixed point if the following conditions are satisfied, respectively.*

**Case (i):** *both* $\mathbf{A}$ *and* $(\mathbf{X} + \mathbf{E})^\top \mathbf{DPE}$ *are positive definite.* **Case (ii):** *both* $\mathbf{A}$ *and* $-\mathbf{X}^\top \mathbf{DPE}$ *are positive definite.* **Case (iii):** $\mathbf{A}$ *is positive definite.*

From the convergence conditions for the three cases in Theorem 4, it is clear that (iii) is the mildest. This is the same condition as that in the normal TD learning without noisy state observations. Note that the case (iii) can be viewed as the SN-MDP setting, whose convergence has been already rigorously analyzed in Section 3. In Section 5, our experiments demonstrate that both expectation-based and distribution RL are more likely to converge in case (iii) compared with case (i) and (ii).

In cases (i) and (ii), the positive definiteness of $\mathbf{X}^\top \mathbf{DPE} + \mathbf{E}^\top \mathbf{DPE}$ and $-\mathbf{X}^\top \mathbf{DPE}$ is crucial. We partition $(\mathbf{X} + \mathbf{E})^\top \mathbf{DPE}$ into $\mathbf{X}^\top \mathbf{DPE} + \mathbf{E}^\top \mathbf{DPE}$, where the first term has the opposite positive definiteness to $-\mathbf{X}^\top \mathbf{DPE}$, and the second term is positive definite (Sutton & Barto, 2018). Based on these observations, we discuss the subtle convergence relationship in cases (i) and (ii):

**(1)** If $-\mathbf{X}^\top \mathbf{DPE}$ is positive definite, which indicates that TD is convergent in case (ii), TD can still converge in case (i) **unless** the positive definiteness of $\mathbf{E}^\top \mathbf{DPE}$ dominates in $\mathbf{X}^\top \mathbf{DPE} + \mathbf{E}^\top \mathbf{DPE}$.

**(2)** If $-\mathbf{X}^\top \mathbf{DPE}$ is negative definite, TD is likely to diverge in case (ii). By contrast, TD will converge in case (i).

In summary, there exists a subtle trade-off of TD convergence in case (i) and (ii) if we approximately ignore the term $\mathbf{E}^\top \mathbf{DPE}$ in case (i). The key of it lies in the positive definiteness of the matrix $\mathbf{X}^\top \mathbf{DPE}$, which heavily depends on the task. In Section 5, we empirically verify that the convergence situations for current and next state observations are normally different. Which situation is superior is heavily dependent on the task.

## E  SENSITIVITY ANALYSIS BY INFLUENCE FUNCTION

Next, we conduct an outlier analysis by the *influence function*, a key facet in the robust statistics (Huber, 2004). The influence function characterizes the effect that the noise in particular observation has on an estimator, and can be utilized to investigate the impact of one particular state observation noise on the training of reinforcement learning algorithms. Specifically, suppose that $F_\epsilon$ is the contaminated distribution function that combines the clear data distribution $F$ and an outlier $x$. The distribution $F_\epsilon$ can be defined as

$$F_\epsilon = (1 - \epsilon)F + \epsilon \delta_x, \tag{31}$$

where $\delta_x$ is a probability measure assigning probability 1 to $x$. Let $\hat{\theta}$ be a regression estimator. The influence function of $\theta$ at $F$, $\psi : \mathcal{X} \to \Gamma$ is defined as

$$\psi_{\hat{\theta}, F}(x) = \lim_{\epsilon \to 0} \frac{\hat{\theta}\left(F_\epsilon(x)\right) - \hat{\theta}(F)}{\epsilon}. \tag{32}$$

Mathematically, the influence function is the Gateaux derivative of $\theta$ at $F$ in the direction $\delta_x$. Owing to the fact that traditional value-based RL algorithms, e.g., DQN (Mnih et al., 2015), can be viewed as a regression problem (Fan et al., 2020), the linear TD approximator also has a strong connection with regression problems. Based on this correlation, in the following Theorem 5, we quantitatively evaluate the influence function of TD learning in the case of linear function approximation.

**Theorem 5.** *(Influence Function Analysis in TD Learning with linear function approximation) Denote* $d_t = \mathbf{x}_t - \gamma \mathbf{x}_{t+1} \in \mathbb{R}^d$, *and* $\mathbf{A} \doteq \mathbb{E}\left[\mathbf{x}_t d_t^\top\right] \in \mathbb{R}^{d \times d}$. *Let* $F_\pi$ *be the data distribution generated from the environment dynamics given a policy* $\pi$. *Consider an outlier pair* $(\mathbf{x}_t, \mathbf{x}_{t+1})$ *with the reward* $R_{t+1}$, *the influence function* $\psi$ *of this pair on the estimator* $\mathbf{w}$ *is derived as*

$$\psi_{\mathbf{w}, F_\pi}(\mathbf{x}_t, \mathbf{x}_{t+1}) = \mathbb{E}(\mathbf{A}^\top \mathbf{A})^{-1} d_t \mathbf{x}_t^\top \mathbf{x}_t (R_{t+1} - d_t^\top \mathbf{w}). \tag{33}$$

Please refer to Appendix G for the proof. Theorem 5 shows the quantitative impact of an outlier pair $(\mathbf{x}_t, \mathbf{x}_{t+1})$ on the learned parameter $\mathbf{w}$. Moreover, a corollary can be immediately obtained to make a precise comparison of the impacts of perturbations on current and next state features.

**Corollary 1.** *Given the same perturbation* $\eta$ *on either current or next state features, i.e.,* $\mathbf{x}_t$, *and* $\mathbf{x}_{t+1}$, *at the step* $t$, *if we approximate* $\eta \eta^\top \mathbf{x}_t$ *and* $\eta \eta^\top \mathbf{w}$ *as* $\mathbf{0}$ *as* $\eta$ *is small enough, the following relationship between the resulting variations of influence function,* $\Delta_{\mathbf{x}_t} \psi$ *and* $\Delta_{\mathbf{x}_{t+1}} \psi$, *holds:*

$$\gamma \Delta_{\mathbf{x}_t} \psi + \Delta_{\mathbf{x}_{t+1}} \psi = 2\gamma d_t \eta \mathbf{x}_t^\top (R_{t+1} - d_t^\top \mathbf{w}). \tag{34}$$

We provide the proof of Corollary 1 in Appendix G. Under this equation, the sensitivity of noises on $\mathbf{x}_t$ and $\mathbf{x}_{t+1}$, measured by $\Delta_{\mathbf{x}_t} \psi$ and $\Delta_{\mathbf{x}_{t+1}} \psi$, present a trade-off relationship as their weighted sum is definite. However, there is not an ordered relationship between $\Delta_{\mathbf{x}_t} \psi$ and $\Delta_{\mathbf{x}_{t+1}} \psi$. In summary, we conclude that the sensitivity of current and next state features against perturbations is normally divergent, and the degree of sensitivity is heavily determined by the task. These conclusions are similar to those we derived in the TD convergence part.

## F   PROOF OF THEOREM 4

*Proof.* To prove the convergence of TD under the noisy states, we use the results from (Borkar & Meyn, 2000) that require the condition about stepsizes $\alpha_t$ holds: $\sum_{t=0}^{\infty} \alpha_t = \infty$ and $\sum_{t=0}^{\infty} \alpha_t^2 = 0$. Based on (Sutton & Barto, 2018), the positive definiteness of $\mathbf{A}$ will determine the TD convergence. For linear TD(0), in the continuing case with $\gamma < 1$, $\mathbf{A}$ can be re-written as:

$$
\begin{aligned}
\mathbf{A} &= \sum_s \mu(s) \sum_a \pi(a|s) \sum_{r,s'} p(r,s'|s,a)\mathbf{x}_t \left(\mathbf{x}_t - \gamma \mathbf{x}_{t+1}\right)^\top \\
&= \sum_s \mu(s) \sum_{s'} p(s'|s)\mathbf{x}_t \left(\mathbf{x}_t - \gamma \mathbf{x}_{t+1}\right)^\top \\
&= \sum_s \mu(s)\mathbf{x}_t(\mathbf{x}_t - \gamma \sum_{s'} p(s'|s)\mathbf{x}_{t+1})^\top \\
&= \mathbf{X}^\top \mathbf{D}\mathbf{X} - \mathbf{X}^\top \mathbf{D}\gamma\mathbf{P}\mathbf{X} \\
&= \mathbf{X}^\top \mathbf{D}(\mathbf{I} - \gamma\mathbf{P})\mathbf{X}
\end{aligned}
\tag{35}
$$

Then we use $\mathbf{A}_t$ to present the convergence matrix in the case (i) where the perturbation vector $\mathbf{e}_t$ is added onto the current state features, i.e., $\mathbf{x}_t \leftarrow \mathbf{x}_t + \mathbf{e}_t$, while we use $\mathbf{A}_{t+1}$ and $\mathbf{A}_{t,t+1}$ to present the counterparts in the case (ii) and (iii), respectively. Based on Eq. 35, in the case (iii), we have:

$$
\begin{aligned}
\mathbf{A}_{t,t+1} &= (\mathbf{X}+\mathbf{E})^\top \mathbf{D}(\mathbf{X}+\mathbf{E}) - (\mathbf{X}+\mathbf{E})^\top \mathbf{D}\gamma\mathbf{P}(\mathbf{X}+\mathbf{E}) \\
&= (\mathbf{X}+\mathbf{E})^\top \mathbf{D}(\mathbf{I}-\gamma\mathbf{P})(\mathbf{X}+\mathbf{E})
\end{aligned}
\tag{36}
$$

From (Sutton & Barto, 2018), we know that the inner matrix $\mathbf{D}(\mathbf{I}-\gamma\mathbf{P})$ is the key to determine the positive definiteness of $\mathbf{A}$. If we assume that $\mathbf{A}$ is positive definite, which also indicates that $\mathbf{D}(\mathbf{I}-\gamma\mathbf{P})$ is positive definite equivalently. As such, $\mathbf{A}_{t,t+1}$ is positive definite automatically, and thus the liner TD would converge to the TD fixed point. Next, in the case (i) we have:

$$
\begin{aligned}
\mathbf{A}_t &= (\mathbf{X}+\mathbf{E})^\top \mathbf{D}(\mathbf{X}+\mathbf{E}) - (\mathbf{X}+\mathbf{E})^\top \mathbf{D}\gamma\mathbf{P}\mathbf{X} \\
&= \mathbf{A} + \mathbf{X}^\top \mathbf{D}\mathbf{E} + \mathbf{E}^\top \mathbf{D}\mathbf{X} + \mathbf{E}^\top \mathbf{D}\mathbf{E} - \mathbf{E}^\top \mathbf{D}\gamma\mathbf{P}\mathbf{X} \\
&= (\mathbf{X}+\mathbf{E})^\top \mathbf{D}(\mathbf{I}-\gamma\mathbf{P})(\mathbf{X}+\mathbf{E}) + (\mathbf{X}+\mathbf{E})^\top \mathbf{D}\gamma\mathbf{P}\mathbf{E} \\
&= \mathbf{A}_{t,t+1} + \gamma(\mathbf{X}+\mathbf{E})^\top \mathbf{D}\mathbf{P}\mathbf{E} \\
&= \mathbf{A}_{t,t+1} + \gamma(\mathbf{X}^\top \mathbf{D}\gamma\mathbf{P}\mathbf{E} + \mathbf{E}^\top \mathbf{D}\gamma\mathbf{P}\mathbf{E})
\end{aligned}
\tag{37}
$$

Similarly, in the case (ii), we can also attain:

$$
\begin{aligned}
\mathbf{A}_{t+1} &= \mathbf{X}^\top \mathbf{D}\mathbf{X} - \mathbf{X}^\top \mathbf{D}\gamma\mathbf{P}(\mathbf{X}+\mathbf{E}) \\
&= \mathbf{A} - \gamma\mathbf{X}^\top \mathbf{D}\mathbf{P}\mathbf{E}
\end{aligned}
\tag{38}
$$

We know that the positive definiteness of $\mathbf{A}$ and $\mathbf{A}_{t,t+1}$ is only determined by the positive definiteness of the inner matrix $\mathbf{D}(\mathbf{I}-\gamma\mathbf{P})$. If we assume the positive definiteness of $\mathbf{A}$, i.e., the positive definiteness of $\mathbf{A}_{t,t+1}$ and $\mathbf{D}(\mathbf{I}-\gamma\mathbf{P})$, as $\gamma > 0$, what we only need to focus on are the positive definiteness of $\mathbf{X}^\top \mathbf{D}\mathbf{P}\mathbf{E}+\mathbf{E}^\top \mathbf{D}\mathbf{P}\mathbf{E}$ and $-\mathbf{X}^\top \mathbf{D}\mathbf{P}\mathbf{E}$. If they are positive definite, TD learning will converge under their cases, respectively.   □

## G   PROOF OF THEOREM 5 AND COROLLARY 1

*Proof.* We combine the proof of Theorem 5 and Corollary 1 together. The TD fixed point $\mathbf{w}_{\text{TD}}$ to the system satisfies $\mathbf{A}\mathbf{w}_{\text{TD}} = \mathbf{b}$. Thus, the TD convergence point, i.e., TD fixed point, can be attained by solving the following regression problem:

$$
\min_{\mathbf{w}} \|\mathbf{b} - \mathbf{A}\mathbf{w}\|^2
\tag{39}
$$

To derive the influence function, consider the contaminated distribution which puts a little more weight on the outlier pair $(\mathbf{x}_t, \mathbf{x}_{t+1})$:

$$
\begin{aligned}
\hat{\mathbf{w}} = \arg\min_{\mathbf{w}}(1-\epsilon)\mathbb{E}[(\mathbf{b} - \mathbf{A}\mathbf{w})^\top (\mathbf{b} - \mathbf{A}\mathbf{w})]+ \\
\epsilon(y_b - x_A^\top \mathbf{w})^\top (y_b - x_A^\top \mathbf{w}),
\end{aligned}
\tag{40}
$$

where $y_b = R_{t+1}\mathbf{x}_t$ and $x_b = d_t \mathbf{x}_t^\top$. We take the first condition:

$$
(1-\epsilon)\mathbb{E}(2\mathbf{A}^\top \mathbf{A}\mathbf{w} - 2\mathbf{A}^\top \mathbf{b}) - 2\epsilon x_A(y_b - x_A^\top \mathbf{w}) = 0.
\tag{41}
$$

Then we arrange this equality and obtain:

$$(1 - \epsilon)\mathbb{E}(\mathbf{A}^\top \mathbf{A} + x_A x_A^\top)\mathbf{w}_\epsilon = (1 - \epsilon)\mathbb{E}(\mathbf{A}^\top \mathbf{b}) + \epsilon x_A y_b. \tag{42}$$

Then we take the gradient on $\epsilon$ and let $\epsilon = 0$, then we have:

$$(-\mathbb{E}(\mathbf{A}^\top \mathbf{A}) + x_A x_A^\top)\mathbf{w}_\epsilon + \mathbb{E}(\mathbf{A}^\top \mathbf{A})\psi_{\mathbf{w}, F_\pi} = -\mathbb{E}(\mathbf{A}^\top \mathbf{b}) \\ + x_A y_b. \tag{43}$$

We know that under the least square estimation, the closed-form solution of $\mathbf{w}_\epsilon$ is $\mathbb{E}(\mathbf{A}^\top \mathbf{A})^{-1}\mathbb{E}(\mathbf{A}^\top \mathbf{b})$. Thus, after the simplicity, we finally attain:

$$\psi_{\mathbf{w}, F_\pi}(\mathbf{x}_t, \mathbf{x}_{t+1}) = \mathbb{E}(\mathbf{A}^\top \mathbf{A})^{-1} x_A (y_b - x_A^\top \mathbf{w}) \\ = \mathbb{E}(\mathbf{A}^\top \mathbf{A})^{-1} d_t \mathbf{x}_t^\top \mathbf{x}_t (R_{t+1} - d_t^\top \mathbf{w}). \tag{44}$$

Next, we prove the Corollary. We only need to focus on the item $d_t \mathbf{x}_t^\top \mathbf{x}_t (R_{t+1} - d_t^\top \mathbf{w})$, which we denote as $\psi_0$. Then we use $\Delta_{x_t}\psi$ and $\Delta_{x_{t+1}}\psi$ to represent the change of $\psi$ after adding perturbations $\eta$ on $\mathbf{x}_t$ and $\mathbf{x}_{t+1}$, respectively. In particular, since we approximate $\eta\eta^\top \mathbf{x}_t$ and $\eta\eta^\top \mathbf{w}$ as $\mathbf{0}$, then we have that the change of influence function for the perturbation $\eta$ on the current state feature $\mathbf{x}_t$:

$$\begin{aligned} \Delta_{x_t}\psi &\approx (d_t + \eta)(\mathbf{x}_t^\top \mathbf{x}_t + 2\eta^\top \mathbf{x}_t)(R_{t+1} - d_t^\top \mathbf{w} - \eta^\top \mathbf{w}) - \psi_0 \\ &\approx -d_t \mathbf{x}_t^\top \mathbf{x}_t \eta^\top \mathbf{w} + 2 d_t \eta^\top \mathbf{x}_t (R_{t+1} - d_t^\top \mathbf{w}) + \eta \cdot \mathbf{x}_t^\top \mathbf{x}_t (R_{t+1} - d_t^\top \mathbf{w}) \\ &= 2 d_t \eta^\top \mathbf{x}_t (R_{t+1} - d_t^\top \mathbf{w}) - \frac{1}{\gamma}(\gamma d_t \mathbf{x}_t^\top \mathbf{x}_t \eta^\top \mathbf{w} - \gamma\eta \mathbf{x}_t^\top \mathbf{x}_t (R_{t+1} - d_t^\top \mathbf{w})). \end{aligned} \tag{45}$$

Then the influence function for the perturbation $\eta$ on the next state feature $\mathbf{x}_{t+1}$ is:

$$\begin{aligned} \Delta_{x_{t+1}}\psi &= (d_t - \gamma\eta)\mathbf{x}_t^\top \mathbf{x}_t (R_{t+1} - d_t^\top \mathbf{w} + \gamma\eta^\top \mathbf{w}) - \psi_0 \\ &\approx \gamma d_t \mathbf{x}_t^\top \mathbf{x}_t \eta^\top \mathbf{w} - \gamma\eta \mathbf{x}_t^\top \mathbf{x}_t (R_{t+1} - d_t^\top \mathbf{w}). \end{aligned} \tag{46}$$

Finally, it is easy to observe that the following relationship holds:

$$\gamma\Delta_{x_t}\psi = 2\gamma d_t \eta \mathbf{x}_t^\top (R_{t+1} - d_t^\top \mathbf{w}) - \Delta_{x_{t+1}}\psi. \tag{47}$$

□

## H    EXPERIMENTAL SETUP

After a linear search, in the QR-DQN, We set $\kappa = 1$ for the Huber quantile loss across all tasks due to its smoothness.

**Cart Pole**    After a linear search, in the QR-DQN, we set the number of quantiles $N$ to be 20, and evaluate both DQN and QR-DQN on 200,000 training iterations.

**Mountain Car**    After a linear search, in the QR-DQN, we set the number of quantiles $N$ to be 2, and evaluate both DQN and QR-DQN on 100,000 training iterations.

**Breakout and Qbert**    After a linear search, in the QR-DQN, we set the number of quantiles $N$ to be 200, and evaluate both DQN and QR-DQN on 12,000,000 training iterations.

## I    SN-MDP SETTING WITH RANDOM NOISES

From Figure 5, we can easily observe that all DQN and QRDQN algorithms under various strength of random state noises converge, although they eventually obtained different average returns. This empirical observation is consistent with our theoretical analysis in Section 3 where both classical and distributional Bellman operators are contractive and thus convergent. In addition, Figure 5 also manifests that the eventual performance that algorithms attained has a decreasing tendency as the perturbation strength, i.e., standard deviation, increases especially for DQN. Remarkably, the final performance of QRDQN is more robust than DQN against different perturbation strength, especially in Mount Car and Breakout games, although both DQN and QRDQN are convergent in this SN-MDP-like setting.

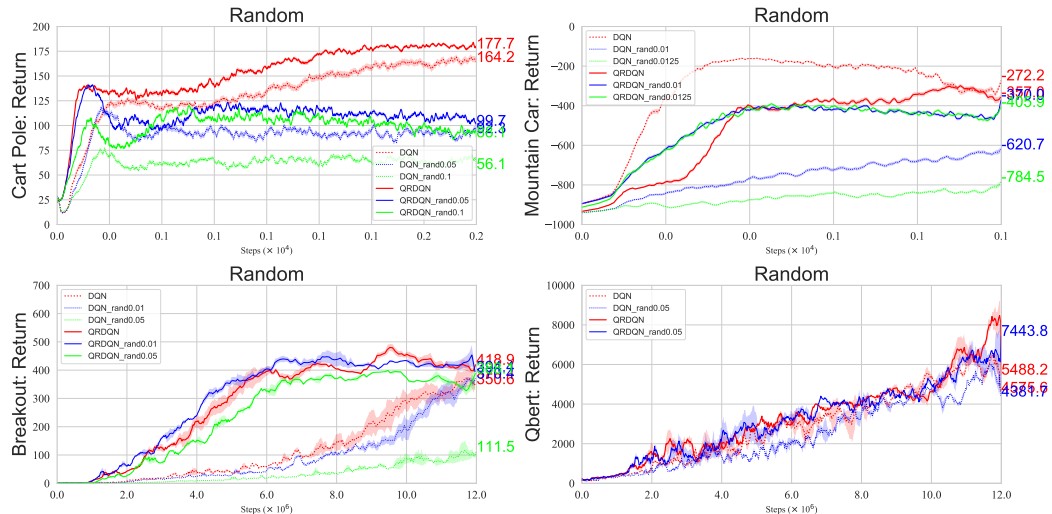

Figure 5: Average returns of DQN and QRDQN against **random** state observation noises across four games. **randX** in the legend indicates random state observations with the standard deviation **X**. Both QRDQN (solid lines) and DQN (dashed lines) converge and the convergence level is determined by the perturbation strength.

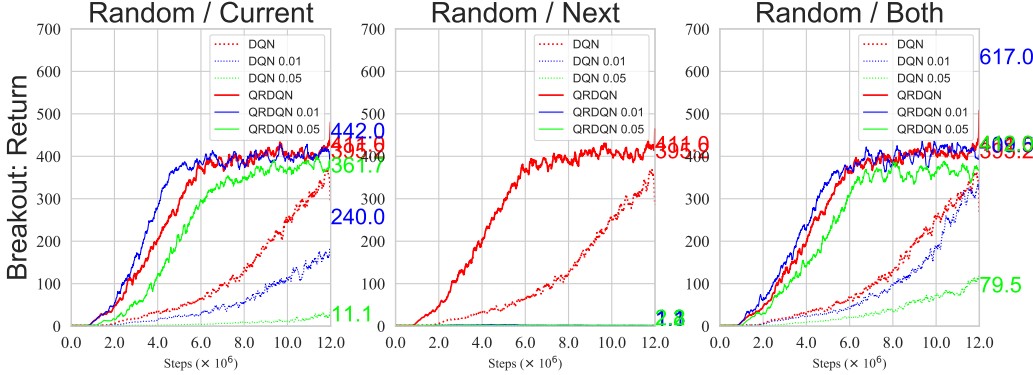

Figure 6: Average returns of DQN and QRDQN against random state observation noises on Breakout environment over 200 runs.

## J    EXTENSION ANALYSIS OF PERTURBATIONS ON CURRENT, NEXT AND BOTH STATE OBSERVATION

We provide the results of robust performance with different perturbation states in Breakout under random and adversarial noisy state observations in Figure 6 and 7, respectively.

In the random state observation setting as shown in Figure 6, it turns out that perturbations on current state observations is more robust than next state by comparing the first and second subfigures given the same perturbation strength. This empirical finding demonstrates the conclusion in Appendix D and E that the sensitivity of current and next states are normally divergent. The result where noises are added on both states seems to be most robust, consistent with the mildest TD convergence condition as shown in Theorem 4.

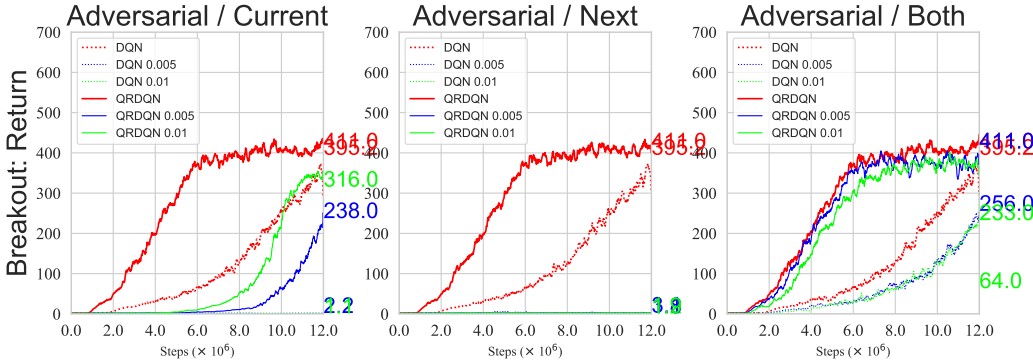

Figure 7: Average returns of DQN and QRDQN against adversarial state observation noises on Breakout environment over 200 runs.

In the adversarial state observation setting as shown in Figure 7, it also turns out that perturbations on current state observations is more robust than next state by comparing the first and second subfigures given the same perturbation strength, and algorithms converge easiest. These empirical results also demonstrates the conclusion we made in Appendix D and E.

