# OpenReview forum: "Exploring the Robustness of Distributional Reinforcement Learning against Noisy State Observations"
_ICLR.cc/2022/Conference — ICLR 2022 Submitted_

### Official Review · Reviewer_DaN6 · 2021-11-02

**Correctness:** 2
**Technical Novelty And Significance:** 2
**Empirical Novelty And Significance:** 1
**Recommendation:** 3
**Confidence:** 5

**Main Review:**

The idea of understanding the distributional robustness of distributional RL seems novel and has not been done before. Other than that, the technical contributions and insights are marginal, unclear, and disconnected:
- Theorem 1 is a marginal extension of (Zhang et al., 2020): they impose a distribution $N$ on a disturbance function $v(s)$
- Function approximation: It was actually a simple linear model with KL loss as the distributional loss for distributional RL, thus the Lipschipts smoothness there is very straightforward and is not really helpful for the real setting of distributional RL. Moreover, In RL, we often consider bounded reward, so if we consider the expected RL with squared loss function, it still enjoys all the properties of KL loss arising from distributional RL
- Disconnected results: How were the analysis in Sections 4.2 and 4.3 related to distributional RL? After the sensitivity analysis, the paper did not give any actionable insights from their analysis except that "the degree of sensitivity is heavily determined by the task" . This conclusion is not helpful and does not need any such sensitivity analysis.
- The paper says that QRDQN is more robust than DQN in MountainCar but Figure 3 tells the opposite. Plus, three experiments are of the same nature; thus I think it is not necessary to split it into three subsections to make them sound comprehensive.
- Section 5.3: "QRDQN eventually achieves similar performance as DQN, although QRDQN significantly reduces the sample efficiency [in Breakout]". From my own experience with this experiment, I don't think so.


**Minor comments**:
- Eq (4) can be simplified
- Section 3.1: "the state space go to infinity": unclear

-

**Summary Of The Paper:**

This paper investigated the adversarial robustness of distributional RL against noisy/adversarial states. They proved the Lipschitz continuity of distributional RL , provided convergence condition for TD update under noisy states, and conducted sensitivity analysis

**Summary Of The Review:**

The idea of exploring adversarial robustness of distributional RL is interesting and novel, but the analysis presented in the paper are marginal, disconnected, and does not support the understanding of the robustness of distributional RL.

===== AFTER REBUTTAL ===
Increase the score to 3

---

> ### Author Response · Authors · 2021-11-14
> **Response**
>
> Thank you for your valuable comments. Below we address all your concerns. Please let us know if there are any further questions.
>
> **Q1.Explanation about Theorem 1.**
>
> We would like to clarify that Theorem 1 is not a marginal extension of Zhang et.al, 2020, as we are focused on the convergence proof of **distributional Bellman operators** rather than the only expectation-based Bellman operators in Zhang et.al, 2020. Our proof procedure is also vastly different as we mainly leverage some properties of Wasserstein distance. Please refer to Appendix for more detailed proof.
>
> **Q2. Concern about Lipchitz continuity.**
>
> To clarify your concern, we additionally add a new subsection Section 4.1 in the revised version based on one paragraph in the original paper and provide a more detailed explanation on this issue. Specifically, the gradient norm of least square loss in the expectation-based RL can be formulated as
>
> $$| U_t-   w_{t}^{\top} {x}_t  |  \cdot  || w_t ||$$
>
> where $U_t$ can be either an unbiased estimate via Monte Carlo method with $U_t=\sum_{k=0}^\infty \gamma^k r_{t+k+1}$, or a biased estimate via TD learning with $U_t=r_{t+1}+\gamma w_{t}^{\top} x_{t+1}$. Even though we have the bounded reward assumption, leading to $U_t  \in [\frac{R_{\text{min}}}{1-\gamma}, \frac{R_{\text{max}}}{1-\gamma}]$, we can not bound the $w_{t}^{\top} x_{t}$  as the adversary can impose arbitrary large perturbations on ${x}_{t}$. Thus, the term $|U_t - w_t^T {x}_t|$ is unbounded. More details are provided in Section 4.1 in the revised version. By contrast, the histogram distributional loss in distribution RL can guarantee a bounded gradient norm regardless of the perturbation strength.
>
> **Q3.Explanation about Section 4.2 and 4.3.**
>
> The TD convergence condition and influence function analysis in Sections 4.2 and 4.3 are based on expectation-based and heuristically applied in the distributional RL in experiments in order to provide us with more insights into the different sensitivity of perturbation on current and next states. This analysis can be useful especially in safe-critical scenarios as we can gain more knowledge when the adversary imposes perturbations on either current or next states in the TD learning.  However, your point also makes sense as these two sections seem not to be strongly related to the main conclusion. Thus, we defer Section 4.2 and 4.3 in the appendix in the revised version.
>
> **Q4.Results in Figure 3.**
>
> Without noisy states, DQN (red dotted line) is superior to QRDQN (red solid line) as QRDQN does not necessarily outperform DQN across all games. However, with noisy states, QRDQN (blue and green solid lines) surpasses DQN (green and blue dotted lines), respectively, especially for current and both state settings, thus showing better training robustness. Therefore, the better robustness of QRDQN when exposing state noises regardless of whether it outperforms DQN or not without state noises, in fact makes our conclusion more compelling.
>
> **Q5.Section 5.3.**
>
> In Figure 4 of Section 5.3, both DQN and QRDQN eventually achieved average returns around 400 in Breakout, which are similar. QRDQN (red solid line) converges much faster than DQN (red dotted line), showing more sample efficiency. Our implementation is based on the distributional RL baselines implemented by Zhang (2018) [1], with 2.6k stars. Also, our result is also similar to that in README.md file in this github repository.
>
> [1] Shangtong Zhang. Modularized implementation of deep RL algorithms in pytorch.  https://github.com/ShangtongZhang/DeepRL, 2018.
>
> **Q6.Minor comments.**
>
> We think Eq.(4) cannot be further simplified without additional assumptions because the policy function $\pi_t$ is general and can be nonlinear. ``The state space goes to infinity'' implies that state space can be arbitrarily large, in which similar analysis can be conducted in the continuous state case.
>
> In summary, it would be much appreciated if you can take all the revisions and responses into your consideration. We are glad to reply to all further questions. Thank you so much indeed.

---

> > ### Comment · Reviewer_DaN6 · 2021-11-28
> > **Thank you for the response**
> >
> > Thank you for the response. I think the re-structure makes the paper look better. I think this direction is interesting and the experimental result in the revised paper looks promising. The authors can explore and improve along this direction to make the contributions more significant. For the present form, I will only increase to "reject" for the revision, for the following reason.
> >
> > The main point of the present paper is to show the robustness of distributional RL to adversarially noisy states. Their main argument is that the KL (distributional) loss is Liptchiz while expected RL has unbounded gradient norm, thus distributional RL is more robust than expected RL (Theorem 2). This is done in a simple linear setting that is highly disconnected from the real setting of distributional RL. I do not think this argument is sufficient for the main claim. For example, by using the setting of their Theorem 2, I can design a loss function that both are trivial and has an arbitrarily small Lipschitz constant (e.g., $\mathcal{L}_{\theta} = \min_i p_i * \epsilon x(s) $, this loss has Lipschitz constant of $\leq \epsilon$). According to their argument, the RL framework with this loss should be more robust than expected RL, but it is not. I think the use of the Lipschitz constant could be the right direction to quantify robustness (as in the vast literature of adversarially robust supervised learning); it just indicates that the simple setting of Theorem 2 does not help much to describe the robustness of distributional RL. In addition, Theorem 2 uses KL but in the experiments, the paper uses quantile loss (i.e., QRDQN), which is also disconnected. The setting of Theorem 2 is too simple to say anything about the robustness of distributional RL.

---

> > > ### Author Response · Authors · 2021-11-28
> > > **Thank you for the further response**
> > >
> > > We thank you for the further valuable comment.
> > >
> > > Firstly, we would say we truly understand your concern as there is indeed still some gap between theoretical analysis and real distributional RL algorithms, including analysis in the linear case but experiments on non-linear, and KL divergence to real Wasserstein distance(quantile loss). However, it is **a rule of thumb** to conduct analysis in linear case and then heuristically extend to the non-linear case to demonstrate the analysis results as in vast RL literature. Moreover, as directly analyzing Wasssertein distance or Quantile loss is very tricky, we focus on the KL divergence distributional RL loss, thus **allowing theoretical analysis**. We argue that this strategy is OK for the theoretical analysis, and we demonstrate the correctness through rigorous experiments, although analysis on quantile loss and the non-linear case would definitely make our conclusion more trustworthy.

---

> ### Author Response · Authors · 2021-11-28
> **Please give us further opinions on our paper**
>
> Dear Reviewer DaN6,
>
> We have replied to your comments. Would you please check whether our efforts are satisfactory and raise your score? There is only one day left. Many thanks!
>
> Authors

---

### Official Review · Reviewer_aTVu · 2021-11-04

**Correctness:** 3
**Technical Novelty And Significance:** 3
**Empirical Novelty And Significance:** 3
**Recommendation:** 6
**Confidence:** 4

**Main Review:**

Strengths:

1. This paper studies the robustness of distributional RL agents, which were never done by previous works. The results are quite promising and show that distributional RL might be more robust than expectation-based RL agents under noises on state observations.

2. The analysis includes both tabular and function approximation case, and an important theorem on the Lipschitz continuity of distributional RL is given.

3. Experiments are relatively comprehensive.

Weaknesses:

1. The proposed SN-MDP seems to be very similar to SA-MDP (Zhang et al., 2020). I understand the analysis under the distributional RL setting is new, but it is not clear to me what is the main difference in terms of SA-MDP formulation. Can you explain?

1. The paper is not well organized and writing can be improved. For example, there is no need to create 4 sub-sections for showing experimental results on 4 models. I think one subsection for random noise and one subsection for adversarial noise can be better. Theorems are not well motivated and the writing is hard to follow, especially in Section 4.2 and 4.3.

Questions:

1. In Figure 3 the DQN lines (dotted red lines) look better than QRDQN, which do not match texts ("QRDQN (solid lines) almost consistently outperforms
DQN"). Is it a plotting error?

2. Is it possible to empirically measure the Lipschitz continuity (Theorem 2), the positive definitiveness of the matrix $X^T DPE$ (Theorem 3), and the influence function (Theorem 4) on a toy example?

**Summary Of The Paper:**

This paper studies the robustness of distributional reinforcement learning, in particular the robustness on state observations, which have been demonstrated in a few papers on adversarial attacks to deep reinforcement learning. Compared to existing works on robust reinforcement learning on state observations, the main difference in this work is that it considers the distributional RL setting and also considers noise during training time. Theoretically, the authors find that distributional RL can be more robust under this setting, via the lens of Lipschitz continuity of the loss function and the influence function. The findings are also verified empirically on 4 benchmarks.

**Summary Of The Review:**

I feel this paper does present an interesting study but my main concern on is its presentation. I feel this paper can become a good paper if the weakness and questions mentioned above can be addressed. I currently rate the paper at the borderline but I am willing to reevaluate the paper based on the authors' response.

---

> ### Author Response · Authors · 2021-11-14
> **Response**
>
> Thank you for appreciating our work. Below we address all your concerns. Please let us know if there are any further questions.
>
> **Q1. Difference between SN-MDP and SA-MDP.**
>
> The main difference between SN-MDP and SA-MDP is that SN-MDP contains more comprehensive state noises, including both random and adversarial noises, while SA-MDP only focuses on the adversarial setting. The difference leads us to consider two kinds of distributional Bellman operators and further prove their contraction and convergence as shown in our appendix, respectively. In addition, we also simultaneously consider both these two kinds of state noises in our experiments, and thus our analysis is more general than SA-MDP. Most importantly, as you said we are focused on the distributional RL setting.
>
> **Q2. Paper organization.**
>
> We agree with your advice. Thus, as suggested by you we revised our paper and split the experiment into three parts, including SN-MDP, random setting and adversarial setting. The last two parts are aligned with your suggestion and we additionally leverage the first SN-MDP part to demonstrate that both expectation-based and distributional RL can convergence, which is consistent with Theorem 1 and 3 we analyzed in Section 3. In addition, we defer the original Section 4.2 and 4.3 into the appendix in order to let our writing easier to follow. We believe under our revisions both theorems and empirical results echo better with each other, and it would be much appreciated if you could take this revision into your reevaluation for our paper.
>
> **Q3. Explanation of Figure 3.**
>
> We would like to clarify that this conclusion is made when the agent observes the noisy state observations (blue and green lines), where we find QRDQN (blue or green solid lines) is almost superior to DQN (blue or green dashed lines), respectively. In Mountain Car, DQN (red dotted line) indeed outperforms QRDQN (red solid line) without state noises, but under the noisy states, QRDQN is still more robust than DQN, making our conclusion more compelling.
>
> **Q4. Empirical measurement on a toy example.**
>
> We guess your question might be caused by some disturbing proof from our theorems, but we would like to clarify that in the linear function approximation case these proofs in Appendix can be rigorously justified. As suggested by you, we can empirically measure these claims in theorems on a toy example in the appendix of the final version to make our conclusion more rigorous.

---

> > ### Comment · Reviewer_aTVu · 2021-11-22
> > **Thank you for the response**
> >
> > Thank you answering my questions. The explanations on Figure 3 is helpful, and the reorganization of the paper also improved the overall quality and the results are connected better.
> >
> > I still feel the SN-SDP is the same as SA-MDP, because the adversary in the SA-MDP can be a random/gaussian noise adversary (random noise can be seen as a weak adversary). But the distributional Bellman operators proposed by this work is indeed novel. I feel the paper can be more acceptable if the author do not overclaim they propose a new SN-SDP model, but claim that they conduct distributional convergence analysis under the noisy adversary setting in SA-MDP. That will definitely make reviewers more comfortable (other reviewers also have the same concern).
> >
> > My overall rating of this paper stays unchanged, and overall I believe the paper is borderline and I am okay to accept it, although I feel the concerns from reviewer DaN6 are also valid and the authors should work on addressing them to get the paper accepted.

---

> > > ### Author Response · Authors · 2021-11-23
> > > **Thank you for the suggestion**
> > >
> > > Thanks for your further suggestions and for appreciating our work again.

---

### Official Review · Reviewer_crGD · 2021-11-04

**Correctness:** 4
**Technical Novelty And Significance:** 4
**Empirical Novelty And Significance:** 2
**Recommendation:** 5
**Confidence:** 2

**Main Review:**

Authors extended State-Adversarial MDP to SN-MDP by considering more general noise generating mechanism. The theoretical contribution of this work seems clear, but I couldn't find out clear motivation and contribution from the experiments. The given experiments seems more focused on whether SN-MDP's Bellman operators and TD learnings can be used for various control problems or not and whether those are aligned with theoretical intuitions. However, I believe authors should have given experiments on the usefulness of this framework (e.g., authors may suggest more practical but simple problems where training observations are noisy as stated in the introduction). Also, only 3 runs are used for each experiments, and all results are reported without standard errors, which means that we cannot evaluate the statistical significance just by using the reported results. Therefore, I believe the empirical results should be refined for this work to be accepted.

**Summary Of The Paper:**

This work presented State-Noisy Markov Decision Process (SN-MDP), where there is a noise generating mechanism (either from the environment noise or from the adversary), and the theoretical properties (such as convergence and contraction) for corresponding (expected) Bellman operator and distributional Bellman operator were proved. The theoretical analysis was done for both tabular and linear funcion approximation settings. Especially in function approximation setting, authors characterized the robustness blessing of distributional RL based on histogram distributional loss and analyzed how the noise factor affects TD learning by using influence function that utilizes the perburbation method. Empirical analysis was done for DQN and QRDQN by varying noise standard deviations and the position of noise (state/successor state or both), which aims to support the authors' intuition coming from their theorems.

**Summary Of The Review:**

Although the theoretical contribution seems clear, I believe we need additional experiments to support the authors' claim.

---

> ### Author Response · Authors · 2021-11-14
> **Response**
>
> Thank you for your valuable suggestions. Below we address all your concerns. Please kindly let us know if there are any further questions.
>
> **Q1. The motivation and contribution from experiments.**
>
> The key we hope to demonstrate in the experiments is that the training robustness of distributional RL is advantageous to expectation-based RL over different noisy settings, including perturbations on current and next states. For instance, QRDQN(solid lines) consistently outperforms DQN (dashed lines) in the same color (the same perturbation strength) across four games and various kinds of state noises.
>
> As suggested by you, we have re-organized the experiment structure into three parts, including the SN-MDP setting, function approximation with random noises, and with adversarial noises. In particular, in the SN-MDP setting (Section 5.1) we show both expectation-based and distributional RL can converge although into different levels, which is aligned with the convergence results in Theorem 1 of Section 3. Moreover, we demonstrate distributional RL enjoys better training robustness across both random and adversarial noises in Section 5.2 and 5.3, which are consistent with the robustness blessing of distributional RL we analyzed in Section 4. Please refer to the revised version for a more detailed explanation, and it would be much appreciated for you to take account of this revision for the final score.
>
> **Q2. Lacking standard deviation of learning curves.**
>
> Thank you for this useful suggestion and we have added shading indicating the standard errors in the learning curves in the revised version, which still exhibits statistical significance. We will appreciate it if you can finally take this revision into the consideration. We can perform algorithms over more runs in the final version.

---

> ### Author Response · Authors · 2021-11-28
> **Please give us further opinions on our papers**
>
> Dear Reviewer crGD,
>
> We have updated our manuscript and replied to your comments. Would you please check whether our efforts are satisfactory and raise your score? There is only one day left. Many thanks!
>
> Authors

---

### Author Response · Authors · 2021-11-14
**Reponse to all reviewers and ACs**

Dear all reviewers and ACs,

As suggested by all reviewers, in the rebuttal period we re-organized the paper structure to let the experimental part be more aligned with the theoretical results. A new version of our paper has been submitted and the original one is also provided in supplementary materials for reference.

In particular, we split the experiments into three new parts, i.e., SN-MNP, function approximation with random noises, and with adversarial noises suggested by **Reviewer aTVu and DaN6**. Moreover, we defer the original Section 4.2 and 4.3 to the appendix to make our writing easier to follow also suggested by **Reviewer aTVu and DaN6**. A new section 4.1 is given adapted from one paragraph from the original 4.1, which provides more details about the vulnerability of least squared loss in expectation-based RL in spite of the bounded reward assumption that **Reviewer DaN6** concerns.  Also, suggested by **Reviewer crGD**, we improved the presentation by providing all learning curves with standard errors, which never hurt the original conclusions.

**We would like to clarify that all the revisions are still within the scope of the original paper without adding new experiments and new conclusions.** We believe this version can be more impressive and it would be much appreciated for all reviewers and ACs to take this revision into consideration. Thank you so much indeed.

Yours faithfully,

Authors

---

### Decision · Program_Chairs · 2022-01-20

**Decision:**

Reject

**Comment:**

Description of paper content:

A mixed theoretical and experimental paper that investigates the robustness of distributional RL to perturbations of state observations as compared to expectation-based value function learning. They provide sufficient conditions for TD’s convergence and prove the Lipschitz continuity of the loss of a histogram-based KL version of distributional RL with respect to the state features, whereas this is not true for expected RL. This continuity indicates a certain robustness of the loss with respect to perturbations of the state. The theory’s tie to experiment is weak in the sense that it is not predictive of the actual performance of any algorithm. The theoretical methods are based on a previously published paper SA-MDP.

Summary of paper discussion:
The reviewers raised concerns about the statistical significance of the experimental results, the clarity and organization of the writing, the novelty of the theoretical setting, and its usefulness for describing a real problem setting. The majority of reviewers rejected the paper and did not lift the scores after the rebuttal.

(I personally wonder if the community would not benefit from conducting some of these kinds of theoretical analyses and experiments on LQR systems rather than Atari (etc.) environments.)